# ADAPTIVE CAMERA SENSOR FOR VISION MODELS

**Eunsu Baek***
Graduate School of Data Science
Seoul National University
Seoul, Republic of Korea
beshu9407@snu.ac.kr

**Sung-Hwan Han**
Department of Computer Science & ngineering
Seogang University
Seoul, Republic of Korea
sunghwan.NaN@gmail.com

**Taesik Gong**†
Department of Computer Science & Engineering
Ulsan National Institute of Science and Technology
Ulsan, Republic of Korea
taesik.gong@unist.ac.kr

**Hyung-Sin Kim**†
Graduate School of Data Science
Seoul National University
Seoul, Republic of Korea
hyungkim@snu.ac.kr

## ABSTRACT

Domain shift remains a persistent challenge in deep-learning-based computer vision, often requiring extensive model modifications or large labeled datasets to address. Inspired by human visual perception, which adjusts input quality through corrective lenses rather than over-training the brain, we propose *Lens*, a novel camera sensor control method that enhances model performance by capturing high-quality images from the model's perspective, rather than relying on traditional human-centric sensor control. *Lens* is lightweight and adapts sensor parameters to specific models and scenes in real-time (i.e., test-time input adaptation). At its core, *Lens* utilizes *VisiT*, a training-free, model-specific quality indicator that evaluates individual unlabeled samples at test time using confidence scores, without additional adaptation costs. To validate *Lens*, we introduce *ImageNet-ES Diverse*, a new benchmark dataset capturing natural perturbations from varying sensor and lighting conditions. Extensive experiments on both ImageNet-ES and our new *ImageNet-ES Diverse* show that *Lens* significantly improves model accuracy across various baseline schemes for sensor control and model modification, while maintaining low latency in image captures. *Lens* effectively compensates for large model size differences and integrates synergistically with model improvement techniques. Our code and dataset are available at github.com/Edw2n/Lens.git.

## 1 INTRODUCTION

Domain shift, the distribution gap between training and test data, is a well-known challenge that degrades the performance of deep-learning-based computer vision models. Existing solutions mainly focus on either model generalization (Hendrycks et al., 2021; 2019; Sohn et al., 2020; Zhou et al., 2023; Ganin et al., 2016; Cherti et al., 2023; Liu et al., 2021b; 2022; Oquab et al., 2023) or model adaptation (French et al., 2017; Sun & Saenko, 2016; Gong et al., 2024; Yuan et al., 2023; Wang et al., 2022b; Youn et al., 2022), which require modifying the model itself. However, they typically necessitate significant changes to the model and access to large, labeled target datasets, making them costly, time-consuming, and impractical for real-time applications on resource-constrained devices.

In contrast, human visual perception operates through a fine-tuned interplay between the eyes (sensors) and the brain (model). The eyes function as precise sensors, capturing visual data, while the brain processes and interprets it. When visual input is compromised, whether by blurriness or glare, the typical response is to improve the quality of the input through corrective lenses, sunglasses, or magnifying lenses, rather than retraining the brain to interpret flawed images better. This analogy highlights that **the model is not all you need**; acquiring high-quality images through camera sensors is essential to mitigate covariate shifts and improve visual perception (i.e., test-time input adaptation).

---

\* edw2n.github.io.
† Corresponding authors.

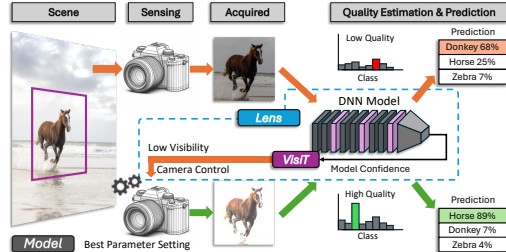

Figure 1: The concept of *Lens*: *Lens* mimics the human vision system, where eyesight quality can be improved through visual sensor control, such as glasses. It leverages sensor parameter adjustments to acquire higher-quality images, thereby enhancing model accuracy.

Despite existing sensor controls like auto-exposure, which are optimized for human perception, we argue that camera sensor control designed for high-quality image acquisition to improve *model perception* requires a fundamentally different approach. Furthermore, in dynamic environments and on resource-constrained devices, sensor control mechanisms must be able to quickly adapt to varying scenes. To address these issues, we introduce *Lens* (Figure 1), a novel adaptive sensor control system that captures high-quality images robust to real-world perturbations. The core idea of *Lens* is to identify optimal sensor parameters that allow the target neural network to better discriminate between objects, akin to adjusting a pair of glasses for clear vision. *Lens* achieves this by leveraging *VisiT* (Vision Test for neural networks), a training-free, model-specific quality indicator that operates on individual unlabeled samples at test time without additional adaptation costs. *VisiT* assesses data quality based on confidence scores tailored to the target model, ensuring high-quality data without the need for extensive retraining or data collection. By acquiring the most discriminative images for the target model, *Lens* significantly boosts model accuracy without requiring model modification.

To demonstrate the effectiveness of *Lens* in realistic sensor control environments, we construct a testbed *ES-Studio Diverse*, where images are captured using a physical camera with varying sensor parameters and light conditions. Using this setup, we create a new dataset called *ImageNet-ES Diverse*, including 192,000 images that capture diverse natural covariate shifts via variations in sensor and light settings, based on 1,000 samples from TinyImageNet (Le & Yang, 2015).

As the first in-depth study on model-centric sensor control, we evaluate *Lens* across two benchmarks – ImageNet-ES (Baek et al., 2024) and our newly created *ImageNet-ES Diverse* – using multiple model architectures. We compare *Lens* against various baselines, including human-targeted or random sensor control methods, domain generalization techniques, and lightweight test-time adaptation (TTA) methods. Our results show that *Lens* with *VisiT* significantly outperforms these methods, improving accuracy by up to **47.58%** while effectively reducing image capture time to only **0.16 seconds**, making it fast enough for real-time operation. The effect of sensor control even compensates for a model size difference of **up to 50×**. Additionally, an ablation study on the quality estimator shows that *VisiT* outperforms state-of-the-art out-of-distribution (OOD) scoring methods, validating confidence scores as an effective quality proxy. Our qualitative analysis further supports these findings with visual insights. The results verify the potential of the new regime: **test-time input adaptation**.

Our key contributions are as follows:

- **Lens**: We introduce *Lens*, a **simple yet effective** adaptive sensor control method that evaluates image quality from the model's perspective and optimizes camera parameters to improve accuracy.
- **ViSiT (Vision Test for neural networks)**: *Lens* adopts *VisiT*, a training-free, model-aware quality indicator that operates on individual unlabeled samples at test time. As the first attempt of its kind, *VisiT* estimates data quality based on confidence scores for **generalizablility and simplicty**.
- **CSAs (Candidate Selection Algorithms)**: We propose CSAs to balance real-time adaptation and accuracy improvements, enabling *Lens*'s efficient operation under practical constraints.
- **ImageNet-ES Diverse**: We release *ImageNet-ES Diverse*, a new benchmark dataset containing 192,000 images that capture natural covariate shifts through varying sensor and lighting conditions.
- **Insights and Findings**: Our extensive experiments not only highlight the superiority of *Lens* but also reveal valuable insights for future research: (1) Sensor control significantly improves accuracy without model modification. (2) Sensor control synergistically integrates with model improvement techniques. (3) Sensor control must be tailored in a model- and scene-specific manner. (4) High-quality images for model perception differ from those optimized for human vision.

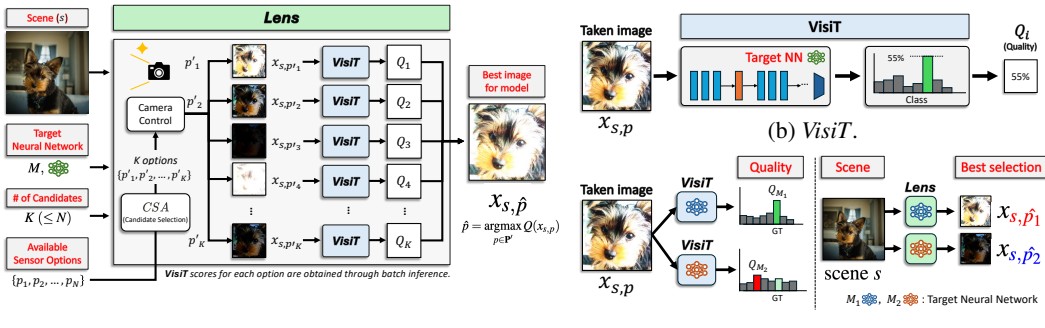

(a) *Lens*.

(b) *VisiT*.

(c) Model-specific design of *VisiT* and *Lens*.

Figure 2: Workflow of *Lens*. *Lens* is a post-hoc, adaptive, and camera-agnostic sensor control system that dynamically responds to scene characteristics while accounting for model- and scene-specific manners based on *VisiT* scores to provide optimal image quality for neural networks.

## 2 RELATED WORK

### 2.1 MODEL IMPROVEMENT: HANDLING DOMAIN-SHIFTED INPUT DATA

Frequent domain shifts pose a significant challenge when deploying neural networks in dynamic real-world environments. Although traditional studies have aimed to improve a model's generalizability or adaptability, these methods place a computational burden, particularly for resource-constrained devices operating in real-time applications. **Domain generalization** techniques (Hendrycks et al., 2021; 2019; Sohn et al., 2020; Zhou et al., 2023; Cherti et al., 2023; Liu et al., 2021b; 2022; Oquab et al., 2023) aim to train models to handle diverse data distributions, but typically results in significantly larger and more complex models. **Domain adaptation** approaches (Ganin et al., 2016; French et al., 2017; Sun & Saenko, 2016) adapt models to a specific target domain, which necessitates frequent retraining and the collection of substantial amounts of labeled target data.

To address the need for lightweight, real-time adaptation without the cost of labeling, **test-time adaptation (TTA)** (Nado et al., 2020; Schneider et al., 2020; Wang et al., 2021) methods have been developed, allowing models to adjust to new domains using a small amount of unlabeled target data with unsupervised objectives. However, these lightweight TTA methods can lead to model collapse when faced with rapidly changing environments.

Lastly, a fundamental limitation of these model-centric techniques is their inability to address the data acquisition process itself. They struggle to cope with severe domain shifts that stem from low-quality data, such as images captured in over-exposed or low-light conditions (Baek et al., 2024).

### 2.2 INPUT DATA IMPROVEMENT: MITIGATING DOMAIN SHIFTS

To address domain shifts through improved data quality, camera sensor control has recently gained attention. Unlike traditional camera auto-exposure methods designed for human perception (Kuno et al., 1998; Liang et al., 2007), this new research focuses on optimizing sensor inputs specifically for deep-learning models. However, the absence of suitable benchmark datasets led early work to rely on camera sensor simulation (Paul et al., 2023), which falls short in generalizing to real-world domain shifts. Although some research has explored the control of physical camera sensors (Odinaev et al., 2023; Onzon et al., 2021), these efforts have been limited to highly-constrained environments with only a narrow range of exposure options.

To overcome these shortcomings, the ImageNet-ES dataset (Baek et al., 2024) was introduced, capturing domain shifts in real-world conditions by employing a physical camera with varying sensor parameters, such as ISO, shutter speed, and aperture. While the ImageNet-ES dataset demonstrates the potential of sensor control in addressing covariate shifts, identifying the optimal sensor parameters for specific models remains an open challenge. Furthermore, additional benchmark datasets are needed to enhance the generalizability of emerging control mechanisms. To the best of our knowledge, this work offers the first comprehensive exploration on camera sensor control using realistic benchmarks, including ImageNet-ES and our newly introduced *ImageNet-ES Diverse* dataset.

## 3 *Lens*: ADAPTIVE GLASSES FOR VISION MODELS

We introduce *Lens*, a post-hoc, adaptive, and camera-agnostic sensor control system for neural networks, designed to adaptively respond to dynamic scene characteristics. The key idea behind *Lens* is to identify the optimal sensor control parameters that capture images in a way that enhances the target model's ability to discriminate features–both in a model-specific and scene-specific manner– akin to adjusting a pair of prescription glasses to provide clear vision tailored to an individual's needs and environment. By **focusing solely on sensor parameter adjustments** and avoiding any modifications to the model itself, *Lens* prevents model collapse and catastrophic forgetting, ensuring reliable performance across varying domains. Moreover, it is lightweight and efficient in terms of both computation and memory. To achieve this, we propose *VisiT* (Vision Test for Neural Networks), a lightweight vision tester integrated into *Lens* that evaluates whether the images captured by the camera sensor are optimally suited for the target model and scene. *VisiT* operates during test time on individual unlabeled samples without modifying the target model.

### 3.1 OVERALL FRAMEWORK

Figure 2a illustrates the overall framework of *Lens*, which operates with a target neural network $M$ that supports batch inference and a camera sensor equipped with a set of $N$ available parameter options, $\mathbf{P} = \{p_1, p_2, \ldots, p_N\}$. Let $x_{s,p}$ represent the image captured by the camera from a target scene $s$ using a sensor parameter option $p$. The goal of *Lens* is to select the optimal sensor parameter $\hat{p}$ such that the captured image $x_{s,\hat{p}}$ maximizes the accuracy of the target model's interpretation of the scene $s$. Let $Q(x_{s,p}; M)$ denote the quality estimate for image $x_{s,p}$ in the context of model $M$. The optimal parameter option $\hat{p}$, as selected by *Lens*, can be represented as:

$$\hat{p} = \arg\max_{p \in \mathbf{P}} Q(x_{s,p}; M)$$

**Model- and Scene-Specific Sensor Control.** *Lens* adaptively selects the optimal sensor parameter $\hat{p}$ for each model and scene in real-time, rather than relying on a globally fixed parameter determined through offline training for all models and/or scenes. The key insight is that different models have distinct ways of extracting and prioritizing features for scene interpretation. As shown in Figure 2c, two different models can perceive the same captured image differently (left side of the figure), leading to different optimal parameters for each model, even for the same scene (right side of the figure). Similarly, even with a fixed model, each scene contains unique features that are crucial for accurate prediction (further discussed in Appendix D). As a result, the optimal sensor parameter is likely to vary for each specific combination of model and scene (Baek et al. (2024)).

***VisiT* (Lightweight Vision Test for neural networks).** *Lens* incorporates *VisiT* (Figure 2b) to estimate $Q(x_{s,p}; M)$, which represents the quality of an unlabeled captured image $x_{s,p}$ when interpreted by the target model $M$. *VisiT* is designed for real-time applications, operating as a lightweight and training-free module at test time, providing model-specific quality estimates for unlabeled images. To achieve the design goal, it is essential to determine an appropriate metric as a proxy for image quality. Specifically, we utilize the model's **confidence score** for its prediction on the image $x_{s,p}$ as a simple yet effective proxy for image quality, which will be further discussed in Section 3.2.

**CSA (Candidate Selection Algorithm).** The latency of *Lens* in selecting the optimal parameter highly depends on the camera sensor's latency to capture multiple images for different candidate parameter options. While capturing and evaluating images for all $N$ available parameter options would provide the highest accuracy, it introduces significant latency for a single scene prediction, which is undesirable for real-time operation. To address the issue, *Lens* uses CSA to select a subset of the full parameter set $\mathbf{P}$, denoted as $\mathbf{P}' = \{p'_1, \ldots, p'_K\}$, as the candidate options. The number of candidate options, $K(\leq N)$, can be determined based on the system's need to balance temporal overhead with accuracy. Note that, since *Lens* operates with batch inference, capturing multiple images doesn't incur additional inference costs.

A crucial aspect of CSA is minimizing capture latency without sacrificing accuracy when selecting $K$ candidate options. For example, sensor parameters like shutter speed significantly impact the capture time. Therefore, within the same time budget, it may (or may not) be more beneficial to prioritize multiple high-shutter-speed options over a single low-shutter-speed option, depending on the specific target scene and model. We explore this trade-off by implementing and evaluating several simple CSAs their performance in Section 5 and discuss their camera-agnostic properties in Appendix A.1.

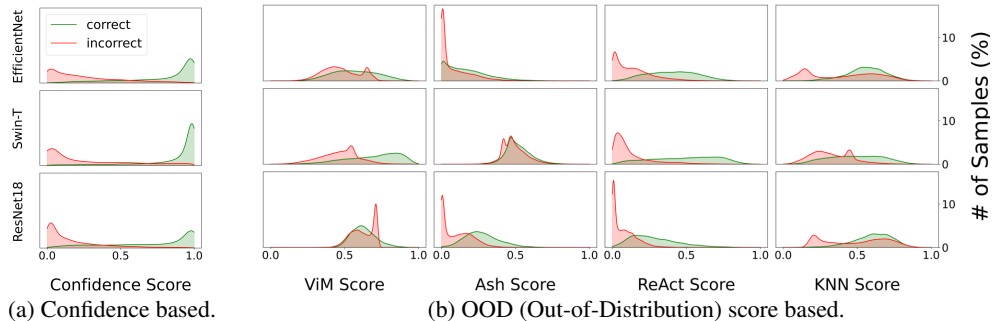

Figure 3: Quality indicators as proxies for image quality assessment: Each score is normalized between 0 to 1.

## 3.2 *VisiT*: LIGHTWEIGHT VISION TEST FOR NEURAL NETWORKS

In this subsection, we provide a detailed description of *VisiT*, the real-time image quality estimator for unlabeled test-time data. The key requirements for *VisiT* design are: (1) Alignment with correctness: The quality estimator must reliably indicate whether the model can accurately predict the sample. (2) Label-free operation: It must function with unlabeled data provided during test time (3) Single-sample assessment: The estimator should be capable of evaluating each image sample independently and immediately. (4) Lightweight operation: It should involve minimal computational overhead, ensuring seamless integration into sensor control pipeline.

**Confidence as a Proxy for Image Quality Assessment.** We propose using the *confidence score* as a simple yet effective proxy. For a sample image $x$ and target model $M$, it is defined as:

$$Confidence(x; M) = \max_{c \in \mathbf{C}} \text{Softmax}(f_M(x))_c$$

where $\mathbf{C}$ is the set of all possible classes, and $f_M(x)$ represents the output logits of the model $M$ before applying the softmax function. The confidence score reflects how certain a model is about its predictions and has been widely used in tasks such as pseudo-labeling, consistency regularization, and high-quality image selection in semi- and self-supervised learning (Oliver et al., 2018; Sajjadi et al., 2016; Sohn et al., 2020; Lee et al., 2013; Cui et al., 2022; Chen et al., 2020; Xie et al., 2020). It is particularly well-suited for real-time applications, as it requires only inference on a sample without incurring additional computational overhead, such as training.

**Correlation between Proxies and Image Quality.** We conducted an experiment to evaluate the correlation between various proxies and image quality under real-world covariate shifts, using the ImageNet-ES validation dataset (Baek et al., 2024) (details in Appendix E.2). We compared our confidence score with out-of-distribution (OOD) scores, commonly used to identify OOD samples, across three models: EfficientNet (Tan & Le, 2019b), Swin-T (Liu et al., 2021b), and ResNet18 (He et al., 2016). The OOD scores were sourced from four state-of-the-art methods: ViM (Wang et al., 2022a), ASH (Djurisic et al., 2023), ReAct (Sun et al., 2021), and KNN (Sun et al., 2022)).

As shown in Figure 3, OOD scores tend to overlap between correct and incorrect samples across all OOD techniques and models, suggesting that OOD scores are not always reliable indicators of image quality. This discrepancy arises because OOD scores are primarily designed to detect semantic shifts (high-level features), but are less effective in identifying covariate shifts, which reflect variations in low-level features. In contrast, samples with higher confidence scores have a greater likelihood of being correct, while those with lower confidence scores are more likely to be incorrect. These results underscore the effectiveness of confidence scores as a reliable proxy for image quality.

## 4 *ImageNet-ES Diverse*: A NEW REAL-WORLD BENCHMARK

*Lens* improves image quality by dynamically controlling camera sensor settings, such as ISO, shutter speed, and aperture, to optimize environmental light for each scene. The quality of an image is significantly influenced by the amount and distribution of light within a scene, which depends on both the characteristics of the objects and the surrounding environment. Therefore, it is essential to evaluate the robustness of *Lens* across various scene characteristics. While the recent ImageNet-ES (Baek et al., 2024) dataset captures real-world scenes with varying sensor parameters, it is limited to only two lighting conditions. Furthermore, as it features images displayed on a screen – representing *light-emitting objects* (e.g., traffic lights) – the impact of ambient light conditions can be restricted.

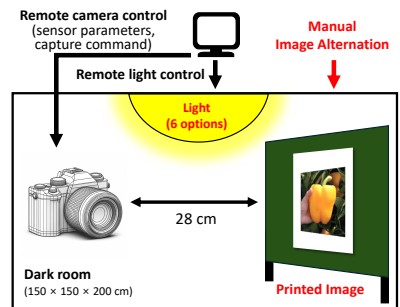

(a) *ES-Studio Diverse*.

(b) Specifics of data collection scheme.

| Considerations | | Descriptions |
|---|---|---|
| Dataset | | Test |
| Original samples | | 1,000 (5 samples / class) |
| Light | | 6 options, L1-L7 (w/o L5) |
| Camera sensor | Auto Exposure | Manual |
| Num. shots | 5 | 27 |
| ISO | Auto | 250 / 2000 / 16000 |
| Shutter speed | Auto | (1/4″) / (1/60″) / (1/1000″) |
| Aperture | Auto | f5.0 / f9.0 / f16 |
| Captured images | 30,000 | 162,000 |

Other collection settings are the same as in the testset of ImageNet-ES Baek et al. (2024), except for the lighting settings.

(c) Light options.

| Light Options | Left | Right |
|---|---|---|
| L1 * | 255 | 255 |
| L2 | 127 | 127 |
| L3 | 255 | 0 |
| L4 | 0 | 255 |
| L6 | 127 | 0 |
| L7 | 0 | 127 |
| L5 * (excluded) | 0 | 0 |

∗: Included in ImageNet-ES (Baek et al., 2024).

Figure 4: Environment and sensor specifics of *ImageNet-ES Diverse*.

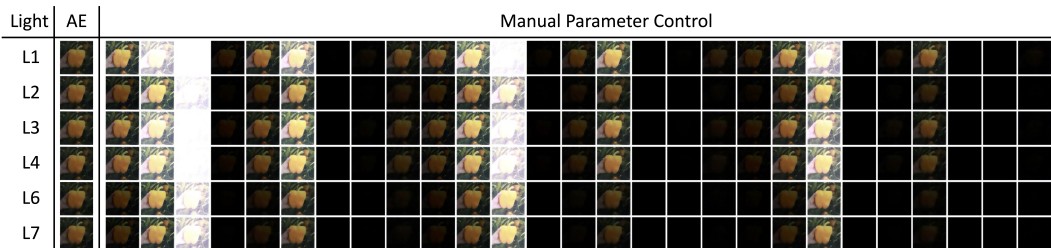

Figure 5: Representative examples of our *ImageNet-ES Diverse* dataset.

To rigorously evaluate *Lens*, a new benchmark dataset is necessary to complement ImageNet-ES and effectively capture the impact of diverse environmental perturbations. To this end, we developed *ImageNet-ES Diverse*, a more versatile dataset with 192,000 samples of *non-illuminous objects* taken with a physical camera on a customized testbed called *ES-Studio Diverse* (Figure 4a). This dataset includes various sensor parameter settings (Figure 4b) and a broader range of lighting conditions (Figure 4c). As illustrated in Figure 5, *ImageNet-ES Diverse* unveils how sensor control interacts with diverse scene characteristics, valuable not only for *Lens* evaluation but also for future research exploring the effects of sensor settings and light conditions. Further details are in Appendix C.

## 5 EXPERIMENTS

We design experiments to evaluate the impact of *Lens*, which is the first approach to introduce **model-and scene-specific sensor control**, in comparison to traditional model-adjustment solutions that completely overlook image capture pipelines and focus solely on over-training for optimizing prediction accuracy under real-world perturbations. Our experiments are conducted across various model architectures, including widely used methods for domain generalization and test-time adaptation.

**Datasets.** We utilize the test sets of **ImageNet-ES** (Baek et al., 2024) and our new *ImageNet-ES Diverse*, both derived from Tiny ImageNet (Le & Yang, 2015) (TIN). These datasets encompass extensive natural perturbations in both environmental and sensor domains including 27 manual controls and 5 auto-exposure shots. ImageNet-ES focuses on Luminous objects, while *ImageNet-ES Diverse* features non-luminous objects, allowing them to complement each other effectively. This diversity allows us to validate our approach across a wide range of real-world covariate shifts. More details about each dataset are in Appendix E.1.

**Baselines and Oracles in the Image Acquisition Pipeline.** For performance comparison, we consider two baselines and two oracles within the data acquisition pipeline. The first baseline, Auto-Exposure (**AE**), is a commonly used sensor control designed to optimize images for human perception, though not necessarily for computer vision models. The second baseline, called **Random**, randomly selects parameter settings, and we calculate its performance as the average over all available options. To explore the potential of model- and scene-specific parameter control, we introduce two oracles: Oracle-Specific (**Oracle-S**) and Oracle-Fixed (**Oracle-F**). Oracle-S ideally selects the best sensor parameter for each sample and model, representing the upper bound for *Lens*. Oracle-F, on the other hand, serves as the upper bound for fixed parameter settings, without considering model-scene interactions. The best global parameter option for Oracle-F is selected based on the average accuracy across all models in Table 1 and all scenes in both datasets.

Table 1: Accuracy comparison among the baselines and *Lens* with various models.

| Model | Num. Params | Pretraining Dataset | DG method | IN | ImageNet-ES (Baek et al., 2024) | | | | | ImageNet-ES Diverse (new) | | | | |
|---|---|---|---|---|---|---|---|---|---|---|---|---|---|---|
| | | | | | Oracle | | Naive control | | *Lens* | Oracle | | Naive control | | *Lens* |
| | | | | | S | F | AE | Random | (Ours) | S | F | AE | Random | (Ours) |
| ResNet-50 (He et al., 2016) | 26M | IN-1K | - | 86.0 | 92.4 | 49.5 | 32.1 | 50.4 | **78.6** | 63.1 | 38.2 | 17.6 | 12.0 | **43.3** |
| | | IN-21K | DeepAugment* +AugMix† | 87.0 | 93.0 | 66.9 | 53.2 | 61.4 | **83.1** | 80.2 | 64.1 | 36.2 | 23.6 | **65.1** |
| ResNet-152 (He et al., 2016) | 60M | IN-1K | - | 87.8 | 93.6 | 58.2 | 41.1 | 54.3 | **81.1** | 69.2 | 43.4 | 21.9 | 14.2 | **48.8** |
| EfficientNet-B0 (Tan & Le, 2019b) | 5M | IN-1K | - | 88.2 | 94.1 | 68.2 | 51.8 | 58.3 | **81.2** | 66.9 | 42.2 | 21.8 | 14.0 | **45.9** |
| EfficientNet-B3 (Tan & Le, 2019b) | 12M | IN-1K | - | 88.1 | 94.8 | 73.9 | 62.0 | 66.3 | **83.5** | 75.8 | 57.2 | 33.6 | 21.4 | **55.7** |
| SwinV2-T (Liu et al., 2022) | 28M | IN-1K | - | 90.6 | 95.1 | 70.4 | 54.1 | 63.1 | **82.6** | 71.5 | 50.8 | 26.5 | 16.9 | **50.3** |
| SwinV2-S (Liu et al., 2022) | 50M | IN-1K | - | 91.7 | 95.4 | 74.4 | 59.9 | 65.5 | **84.5** | 75.3 | 53.9 | 30.8 | 18.9 | **55.6** |
| SwinV2-B (Liu et al., 2022) | 88M | IN-1K | - | 91.9 | 95.3 | 75.3 | 60.0 | 65.5 | **85.3** | 74.6 | 53.6 | 30.8 | 18.5 | **55.3** |
| OpenCLIP-b (Cherti et al., 2023) | 87M | LAION-2B | Text-guided pretrain | 94.3 | 97.4 | 81.4 | 66.1 | 71.3 | **90.9** | 82.7 | 66.4 | 38.8 | 24.5 | **67.6** |
| OpenCLIP-h (Cherti et al., 2023) | 632M | LAION-2B | | 94.9 | 98.5 | 87.1 | 79.0 | 77.6 | **93.0** | 87.9 | 74.6 | 45.5 | 29.3 | **74.4** |
| DINOv2-b (Oquab et al., 2023) | 90M | LVD-142M | Dataset curation | 93.6 | 97.6 | 85.1 | 74.5 | 73.9 | **90.6** | 87.5 | 72.6 | 44.5 | 28.3 | **72.9** |
| DINOv2-g (Oquab et al., 2023) | 1.1B | LVD-142M | | 94.7 | 98.0 | 90.7 | 84.3 | 79.8 | **93.1** | 92.8 | 82.5 | 62.8 | 35.3 | **82.9** |
| All models | | | | 90.7 | 95.4 | 73.4 | 59.8 | 65.6 | **85.6** | 77.3 | 58.3 | 34.2 | 21.4 | **59.8** |

∗: (Hendrycks et al., 2021), †: (Hendrycks et al., 2019), IN: ImageNet (Le & Yang, 2015), S: Specific, F: Fixed, AE: Auto exposure, Random: Random selection

## 5.1 GENERALIZABILITY OF *Lens*

We investigate the effectiveness of *Lens* across various models, including representative (He et al., 2016; Liu et al., 2022), lightweight (Tan & Le, 2019b), and foundation (Cherti et al., 2023; Oquab et al., 2023) models. Furthermore, we examine whether *Lens* can be constructively integrated with domain generalization (DG) techniques. Detailed model setups are in Appendix E.3.1.

Table 1 summarizes the results. While Oracle-F selects the best fixed parameter to maximize average accuracy, it still suffers performance drops in many cases, revealing the limitations of using fixed parameters – *no single parameter optimally supports all scenarios*. In contrast, Oracle-S consistently outperforms Oracle-F by large margins and even matches or exceeds performance on ImageNet (IN), the training domain. This highlights the potential of scene- and model-specific sensor control. More importantly, *Lens* consistently boosts model performance compared to AE and Random across both benchmarks and all models, by large margins ranging from **8.71% to 47.58%**. *Lens* also delivers significantly better worst-case performance than Oracle-F, with gains of 29.1% in ImageNet-ES and 5.43% in *ImageNet-ES Diverse*, demonstrating the robustness of adaptive sensor control. These results show the importance of targeting sensor control to the model, rather than human perception, and demonstrate that *Lens* effectively *unlocks* the potential of model-specific adaptive sensor control.

Moreover, *Lens*, without additional pretraining or extra data collection, outperforms the baseline methods even when they are combined with complex DG techniques like DeepAugment (Hendrycks et al., 2021) and AugMix (Hendrycks et al., 2019)), and applied to significantly larger models. For instance, *Lens* on ResNet-50 (He et al., 2016) outperforms baseline controls on DG-applied ResNet-50, and even those on the larger ResNet-152 (He et al., 2016), with gains ranging from **7.09% to 37.48%**. Furthermore, *Lens* on EfficientNet-B3 (Tan & Le, 2019b), with only 12M parameters, surpasses the DG-enhanced OpenCLIP-h (Cherti et al., 2023), a model with 632M parameters, delivering **7.34%** higher accuracy; *Lens* can compensate for a **50× model size difference** through real-time sensor control. Lastly, when combined with DG techniques and larger models, *Lens*'s performance improves further, highlighting its synergistic nature. These findings emphasize the importance of optimizing data acquisition process, rather than focusing solely on model improvements.

## 5.2 REAL-TIME ADAPTATION PERFORMANCE

To assess the real-time adaptability of *Lens*, we compare its performance with lightweight Test-Time Adaptation (TTA) methods, which are designed for real-time model adaptation. Additionally, we analyze the adaptation cost of *Lens*, focusing on the image capturing overhead associated with selected sensor parameter candidates, demonstrating its efficiency in real-time scenarios.

Table 2: Real-time adaptation performance analysis of *Lens* against TTA methods.

| Model | TIN | Environments | Oracle | | Naive control | | Test-Time Adaptation | | | *Lens* (Ours) | | | |
|---|---|---|---|---|---|---|---|---|---|---|---|---|---|
| | | | S | F | AE | Random | BN1 | BN2 | TENT | Full (k=27) | CSA1 (k=6) | CSA2 (k=6) | CSA3 (k=18) |
| ResNet-18 (He et al., 2016) | 80.4 | ImageNet-ES (Baek et al., 2024) | 87.9 | 54.1 | 39.2 | 46.6 | 30.7 | 34.2 | 32.0 | **73.8** (2.41sec) | 73.4 (0.53sec) | 72.6 (0.53sec) | 73.7 (0.16sec) |
| | | *ImageNet-ES Diverse* | 52.6 | 32.5 | 13.1 | 9.2 | 15.8 | 20.0 | 16.0 | **34.6** (2.41sec) | 26.9 (0.53sec) | 27.4 (0.53sec) | 25.1 (0.16sec) |
| EfficientNet-B0 (Tan & Le, 2019b) | 84.9 | ImageNet-ES (Baek et al., 2024) | 92.3 | 61.2 | 42.6 | 51.2 | 31.9 | 41.7 | 42.6 | **77.8** (2.41sec) | 76.2 (0.53sec) | 77.4 (0.53sec) | 78.6 (0.16sec) |
| | | *ImageNet-ES Diverse* | 60.5 | 38.5 | 19.9 | 11.7 | 15.0 | 17.5 | 15.6 | **39.6** (2.41sec) | 32.4 (0.53sec) | 32.9 (0.53sec) | 31.4 (0.16sec) |

TIN: Tiny-ImageNet (Le & Yang, 2015), AE: Auto Exposure, Random: Random Selection, S: Specific, F: Fixed
BN1: (Nado et al., 2020), BN2: (Schneider et al., 2020), TENT: (Wang et al., 2021)
CSA1: Random Selection, CSA2: Grid Random Selection, CSA3: Cost-Based Selection

**TTA Baselines and Target Models.** We establish three representative TTA baselines: **BN1** (Prediction-time batch normalization, Nado et al. (2020)), **BN2** (Batch Normalization Adaption, Schneider et al. (2020)), and **TENT** (Wang et al., 2021). These methods are applied to the batch normalization layer and offer minimal computational and memory overhead. We apply these TTA baselines to two lightweight models – **ResNet-18** (He et al., 2016) and **EfficientNet-B0** (Tan & Le, 2019b) – using data acquired via the traditional auto-exposure (AE) method. We then compare their performance against the same models when used with data acquired through *Lens*. Detailed explanations of each TTA method and the deployed models are provided in Appendix E.4.

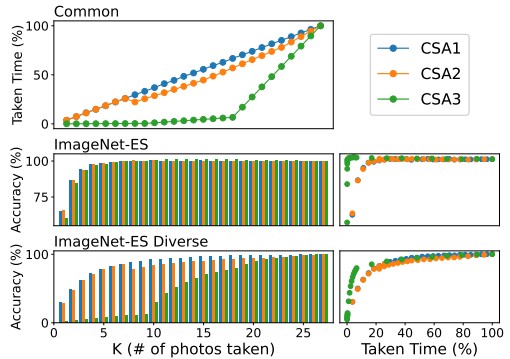

Figure 6: Cost analysis of CSAs (EfficientNet-B0) on ImageNet-ES & *ImageNet-ES Diverse*.

Table 3: Ablations on the Quality Estimator

| | Models | | C1 | C2 | V1 | V2 |
|---|---|---|---|---|---|---|
| | Tiny ImageNet | | 80.4 | 84.9 | 93.7 | 89.3 |
| ImageNet ES | Oracle | S | 87.9 | 92.3 | 96.3 | 95.0 |
| | | F | 54.1 | 61.2 | 81.5 | 74.8 |
| | Naive control | AE | 39.2 | 51.2 | 70.4 | 62.0 |
| | | Random | 46.6 | 42.6 | 71.4 | 66.5 |
| | *Lens* with OOD techniques | ViM | 53.4 | 60.6 | 81.3 | 74.8 |
| | | ReAct | 53.2 | 60.1 | 81.5 | 74.0 |
| | | ASH | 47.2 | 55.9 | 61.2 | 3.2 |
| | | KNN | 53.2 | 60.6 | 81.5 | 74.8 |
| | ***Lens* with VisiT (ours)** | | **73.8** | **77.8** | **89.7** | **85.6** |

Tiny ImageNet (Le & Yang, 2015), ImageNet-ES (Baek et al., 2024)
S: Specific, F: Fixed
AE: Auto exposure, Random: Random selection
C1: ResNet18 (He et al., 2016), C2: EfficientNet-B0 (Tan & Le, 2019b)
V1: Swin-B (Liu et al., 2021b), V2: DeiT (Touvron et al., 2022)

**Candidate Selection Algorithms (CSAs) for *Lens*.** Capturing images for all available parameter options for a scene introduces high latency, so we develop three candidate selection algorithms (CSAs) for *Lens* to enable lightweight, real-time operation. These CSA algorithms consider two key factors: **the number of image captures ($K$)** and **the overall capture time** per scene.

- **CSA1:** A simple method that randomly selects $K$ options from the available options.
- **CSA2:** A grid-based random selection leveraging spatial locality. Observing that parameter settings closer in parameter space often yield similar image qualities, CSA2 divides the parameter space into grids and randomly selects $K$ options from these grids. With 27 available options in our benchmarks (i.e., three options per each of the three parameters), the number of grids becomes $1^3$ for $K = 1$–$7$, $2^3$ for $K = 8$–$26$, and $3^3$ for $K = 27$.
- **CSA3:** This method selects $K$ options with the lowest capture costs, prioritizing settings with shorter shutter speeds, which are the primary contributors to capture latency. If multiple options share the same capture cost, the selection is made randomly.

**Results.** Table 2 shows that *Lens* with full options ($K = 27$) significantly outperforms all TTA baselines across all models and both benchmarks, with gains from **14.6% to 45.9%**. This underscores the superiority of sensor adaptation to model adaptation. Furthermore, the three CSAs for *Lens* drastically reduce capturing time by **93.4%** (to only **0.16 seconds**) or require as few as **6 image captures** while maintaining accuracy. Figure 6 shows detailed interactions between capture time, $K$, and accuracy for EfficientNet-B0 (Tan & Le, 2019b) across both benchmarks, using five random seeds. Note that the correlation between capture time and $K$ is consistent across both benchmarks (marked as "common") because the CSAs are neither model- nor scene-specific, relying instead on non-deterministic selection at the time of capture. While each CSA has a different trade-off between

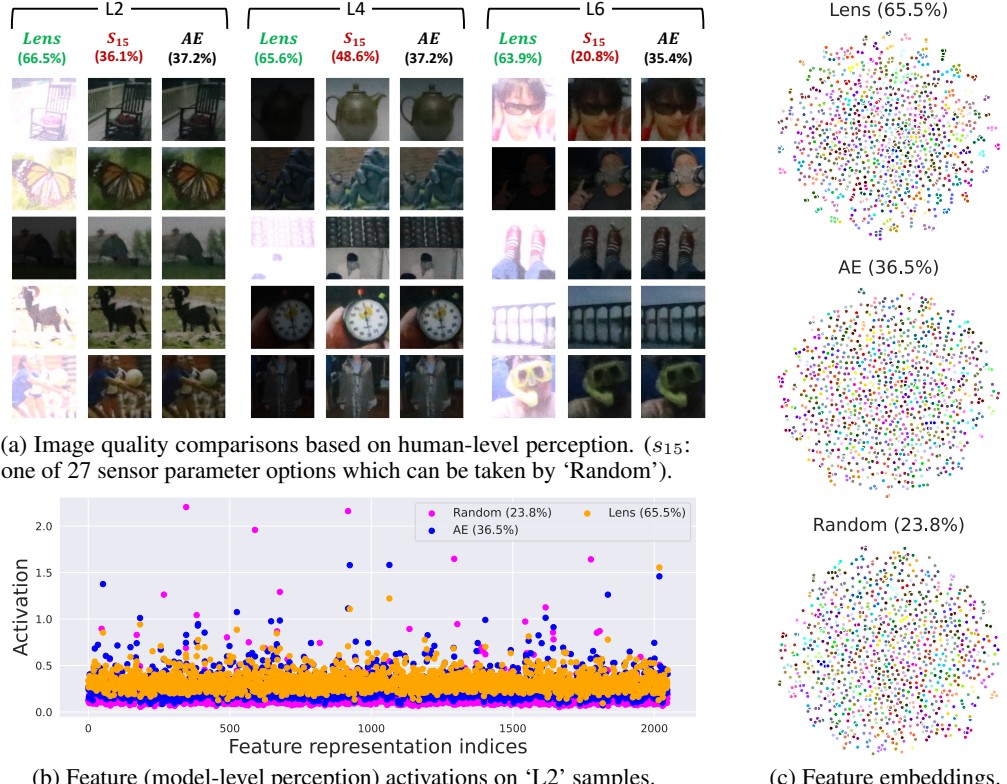

(a) Image quality comparisons based on human-level perception. ($s_{15}$: one of 27 sensor parameter options which can be taken by 'Random').

(b) Feature (model-level perception) activations on 'L2' samples.

(c) Feature embeddings.

Figure 7: Sensing for human vs. sensing for DNN (ResNet-50 (He et al., 2016) augmented with AugMix (Hendrycks et al., 2019) and DeepAug (Hendrycks et al., 2021)) on *ImageNet-ES Diverse*.

$K$ and taken time, all CSAs maintain high accuracy until taken time significantly decreases. These results verify *Lens*'s ability to balance accuracy and efficiency in real-time scenarios.

### 5.3 ABLATION STUDY ON THE QUALITY ESTIMATOR

To investigate the effectiveness of *VisiT*, we replace *VisiT* with four state-of-the-art OOD scoring methods, as introduced in Section 3.2. We evaluate these approaches on the ImageNet-ES dataset across four models: ResNet-18 (He et al., 2016), EfficientNet (Tan & Le, 2019b), Swin-T (Liu et al., 2021b), and DeiT (Touvron et al., 2022). As shown in Table 3, *Lens* integrated with *VisiT* consistently outperforms *Lens* paired with all OOD scoring baselines across every model, achieving an average gain of **20.7%**. This verifies that confidence scores are more reliable than OOD scores for evaluating the image quality from the model's perspective in the face of real-world perturbations.

### 5.4 QUALITATIVE ANALYSIS

**Sensing for Human Vision vs. Model Vision.** Figure 7a highlights the fundamental difference in how humans and neural networks perceive images, using examples from *ImageNet-ES Diverse*. While humans may struggle to discern details in dark or bright images (those selected by *Lens* in L2, L4, and L6), these images lead to better model accuracy (63.9-66.5%). In contrast, models perform poorly (20-48.6%) on images captured using auto-exposure (AE) settings or human-centered settings (S15 in L2, L4, and L6). Figure 7b further emphasizes this perceptual mismatch by showcasing distinct feature activation distributions for sample images under different sensor control methods. Specifically, the images provided by AE and Random settings cause the model to heavily activate certain features (those far from the average) that are treated as marginal for images acquired by *Lens*, which can degrade prediction performance. Moreover, Figure 7c demonstrates that, although *Lens*-acquired images may seem unintuitive from a human perspective, they enable the model to generate feature embeddings—consisting of 1,000 points with 5 points per label and color-coded accordingly—that are more clearly distinguishable between classes compared to those captured with AE and Random settings. These findings highlight the critical need to understand perception differences between humans and neural networks when designing effective sensor control strategies.

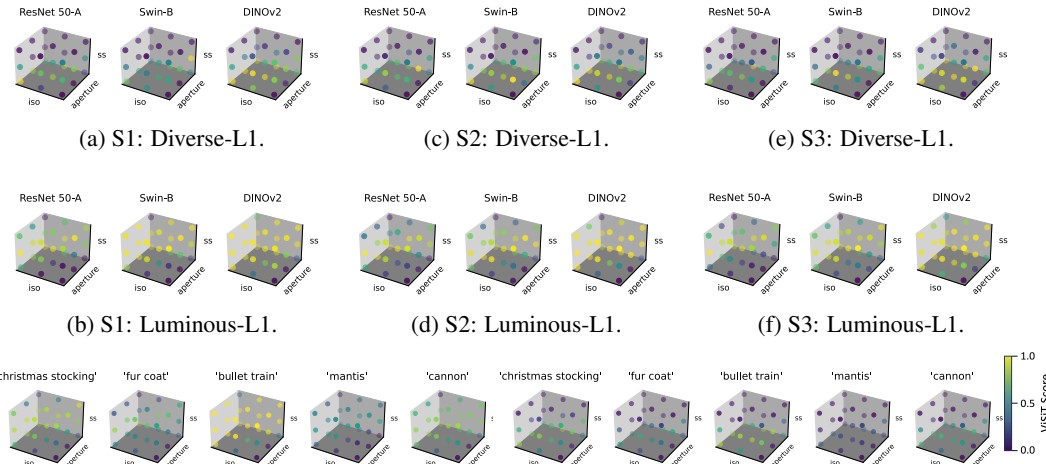

(a) S1: Diverse-L1.    (c) S2: Diverse-L1.    (e) S3: Diverse-L1.

(b) S1: Luminous-L1.   (d) S2: Luminous-L1.   (f) S3: Luminous-L1.

(g) Class-wise Analysis for ResNet50-A. (Luminous) (h) Class-wise Analysis for ResNet50-A. ( *Diverse*)

Figure 8: Model- and scene- specific solution spaces of parameter control in real perturbations. ResNet50-A: ResNet50 (He et al., 2016) + Augmix (Hendrycks et al., 2019) + DeepAugment (Hendrycks et al., 2021), Swin-B (Liu et al., 2022), and DINOv2 (Oquab et al., 2023).

**Solution Space Analysis.** Figure 8 shows the necessity of model- and scene-specific sensor control to handle real-world perturbations. Each grid point is one of the 27 parameter options from ImageNet-ES and *ImageNet-ES Diverse*, color-coded by the *VisiT* score of the image captured with that option. Each subfigure shows the results from three different models, showing that the same parameter setting for an identical sample can yield significantly different quality scores when the model is changed. For example, an optimal parameter for Swin-B (Liu et al., 2022)) may perform poorly for DINOv2 (Oquab et al., 2023) or ResNet18 (He et al., 2016), verifying the need for model-specific control. The figure pairs (8a and 8b), (8c and 8d), and (8e and 8f) represent the same class sample captured under an identical lighting condition but with different object characteristics from "Diverse" scenes in *ImageNet-ES Diverse* and "Luminous" scenes in ImageNet-ES. The column-wise differences between the two datasets emphasize the importance of scene-specific control. With the same sample and L1 setting, fast shutter speeds yield low-quality images in "Diverse" scenes but high-quality images in "Luminous" scenes. Finally, Figures 8g and 8h show that under the same model, lighting conditions, and object characteristics, optimal parameters can vary across different classes. Overall, sensor parameters must be dynamically adjusted based on both model and scene characteristics.

## 6 CONCLUSION

*Lens* is the first method to introduce **model- and scene-specific camera sensor control** inspired by human visual perception; by capturing high-quality images from the model's perspective, *Lens* improves model performance. *Lens* employs *VisiT*, a lightweight, training-free, model-specific quality indicator based on model confidence, which operates on individual unlabeled samples at test time. Evaluations on two benchmarks of real perturbations, including our new dataset *ImageNet-ES Diverse* collected to address previously missing but notable perturbations, demonstrate that *Lens* with *VisiT* improves accuracy by up to **47.58%**, outperforming representative TTA baselines and DG techniques based on naive control. Furthermore, *Lens* shows generalizability across various architectures and can be synergistically combined with all DG methods. By ensuring efficiency in adaptation costs while maintaining performance, *Lens* has the potential for real-time applications (e.g., upcoming embodied AI). Qualitative analysis validates model/scene-specific sensor control's importance, showing its significant impact over DG/TTA, and offering a promising approach for real-world AI adaptability.

**Limitations and Future Work.** While *Lens* presents a novel paradigm of sensing for deep neural networks (test-time input adaptation) with significant potential for adoption in challenging scenarios across various tasks, such as autonomous driving, surveillance, and real-time 3D vision applications, it also opens avenues for further exploration. In this work, model confidence serves as a simple yet effective proxy for image quality assessment, but this can lead to overconfidence, especially in poorly calibrated models. Future work could explore robust quality estimators, synergies with TTA, and improved Candidate Selection Algorithms (CSAs) using model/scene factors and reinforcement learning for resource scheduling. Further discussions are detailed in Appendix A.

## REPRODUCIBILITY STATEMENT

We include the source code in the supplementary material, along with instructions. Detailed information on the experiments, including datasets, scenarios, and hyperparameters, are in Appendix.

## ACKNOWLEDGMENTS

This work was supported in part by Institute of Information communications Technology Planning Evaluation (IITP) grant funded by the Korea government (MSIT) (No. RS-2023-00223530), in part by the National Research Foundation (NRF) of Korea grant funded by the Korea government (MSIT) (No. RS-2023-00222663), and in part by the Creative-Pioneering Researchers Program through Seoul National University.

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

# Appendix

## Adaptive Camera Sensor for Vision Models

## A   FURTHER DISCUSSION

In this section, we discuss future directions of this work, as outlined in Section 6.

### A.1   TOWARDS MORE REALISTIC SCENARIOS

In this study, we utilized ImageNet-ES and *ImageNet-ES Diverse* as real-world perturbations, state-of-the-art Environmental and Sensor (ES) perturbation datasets. These datasets are pioneering in enabling effective evaluation of the impact of sensor control on environmental changes. By leveraging these resources, our work lays a robust foundation for addressing domain shift challenges in more complex and realistic scenarios through sensor control.

#### A.1.1   POTENTIAL OF *Lens* FOR ADAPTATION IN VARIOUS SETTINGS

**More Realistic Datasets.** Extending *Lens* from classification tasks to advanced vision tasks such as semantic segmentation and object detection, and further into applications like autonomous driving or surveillance systems, presents a promising research direction. However, existing datasets lack both sensor control information and the labeled data necessary for these tasks. While ImageNet-ES and *ImageNet-ES Diverse* have facilitated the evaluation of *Lens* for classification, similar datasets tailored to other vision tasks are required. Therefore, the creation and implementation of sensor-controlled datasets for these advanced tasks are crucial for future research on *Lens*. Additionally, to encompass a broader range of realistic scenarios, we intend to collect and integrate more dynamic datasets, including multiple objects and dynamically changing scenes, as well as advanced tasks that incorporate ES perturbations similar to those in ImageNet-ES and *ImageNet-ES Diverse*. This will enable us to validate and enhance the robustness of our methodology against various domain shifts encountered in real-world applications, thereby providing a comprehensive evaluation of our method's resilience and effectiveness across diverse environments.

**Potential to Adaptation on Advanced Vision Tasks.** To showcase *Lens*'s versatility in various vision tasks and its value in collecting dataset containing sensor control factors, we performed a qualitative analysis focusing on two key applications: **Semantic Segmentation** and **Object Detection**. We compared *Lens* with AE (Auto Exposure), a baseline camera sensor control method described in Section 5. The evaluation involved two semantic segmentation models (FCN Long et al. (2015) and DeepLab v3 Chen et al. (2017)) and two object detection models (Faster R-CNN Ren et al. (2016) and SSDLite300 (Liu et al., 2016; Sandler et al., 2018)), representing standard or lightweight architectures. The analysis focused on the 'dog' class, a commonly used category in the training datasets of target models and a superclass in the evaluation datasets. Since these tasks generate multiple outputs, unlike the classification tasks for which *Lens* was initially designed, we adapted *Lens* by modifying the *VisiT* score for each specific task. Detailed experimental setups including the *VisiT* adaptations, are provided in Table 4. As shown in Figures 9 and 10, *Lens* achieves results that closely approximate, and sometimes outperform, those of the original sample (source domain) in most cases for both tasks and all targeted models. In contrast, AE failed to recognize the target class ('dog') in corresponding results. This suggests that *Lens* has significant potential for adaptation to other vision tasks using similar approaches. Furthermore, given that the large models evaluated in Section 5.1 have been consistently improved by our system and share the backbone and datasets of representative Vision-Language Models (VLMs) or curation-based models, we can expect that *Lens* has substantial potential to enhance other VLM models' performance through adaptation. However, the performance of *Lens* varies depending on the customization of the *VisiT* score for each target task, indicating that further elaboration on this aspect represents a promising avenue for future research.

**Generalizability in Heterogeneous Camera Devices** As outlined in the methodology section, while performance values can vary with camera devices, *Lens* operates in a **camera-agnostic manner**, allowing it to be applied regardless of the camera model. *Lens* employs a strategy that selects sensor options to achieve the highest image quality. This strategy remains effective even when camera equipment varies. Specifically, the main modules (*VisiT* and CSAs) used for assessing image quality

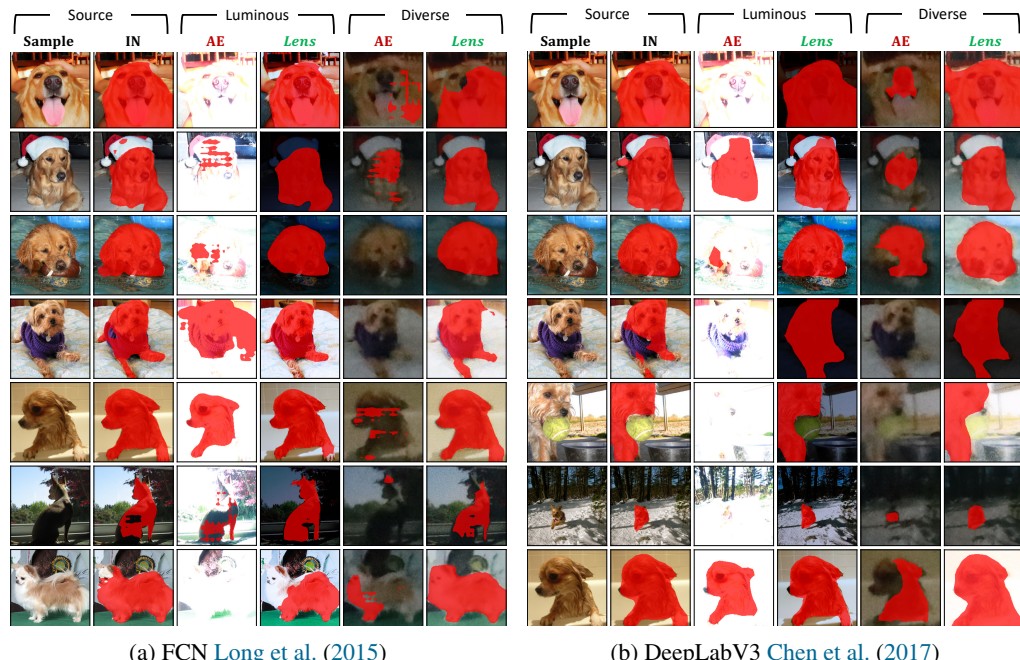

(a) FCN Long et al. (2015)                    (b) DeepLabV3 Chen et al. (2017)

Figure 9: Qualitative Analysis on both Benchmark (Semantic Segmentation)

are camera-agnostic: 1) *VisiT* ensures camera-agnostic functionality by assessing image quality through the confidence scores of images selected by the Camera Selection Algorithm (CSA). and 2) Proposed CSA algorithms in our work are inherently camera-agnostic because they select camera parameter candidates based solely on the provided sensor parameter information, independent of specific camera models. **As long as the necessary information for each CSA algorithm is supplied, they operate regardless of the camera type.** The required information for each proposed CSA algorithm is as follows: i) *Random Selection (CSA1)*: Supported ranges or available sensor parameter options from deployed camera models. ii) *Grid Random Selection (CSA2)*: Grid information of camera parameter ranges based on a specified value of K, derived from camera control specifications. iii) *Cost-Based Selection (CSA3)*: Cost associated with each parameter option across deployed camera models. However, performance may vary depending on specific camera hardware and environmental conditions. Additionally, while it is important to explore methods for more precisely identifying optimal solutions within continuous parameter spaces, it is equally crucial to consider factors such as system latency and adaptability, including training and inference times. To address this, future research should focus on balancing these aspects to facilitate the development of practical and efficient solutions.

### A.1.2    POTENTIAL OF *Lens* FOR MORE CHALLENGING SCENARIOS.

**Addressing Overconfidence.** Although *Lens* has achieved already significant improvements by utilizing confidence scores as quality estimators for sensor control compared to existing baselines, these scores may not be optimal in all scenarios. As highlighted in Section 6, the issue of overconfi-

Table 4: Detailed Settings of Experiments on Other Vision Tasks

| Tasks | Semantic Segmentation | Object Detection |
|---|---|---|
| **Description** | Identify and highlight pixels corresponding to 'dog'. (if the maximum confidence score indicates 'dog') | Detect objects and draw valid bounding boxes. (only for confidence scores >0.6) |
| ***Lens* Adaptation (*VisiT* Score)** | Average of the confidence scores of the highlighted pixels. | Average of the confidence scores of the valid bounding boxes. |
| **Models (backbone)** | FCN Long et al. (2015) (ResNet50), DeepLab v3 Chen et al. (2017) (MobileNet v3) | Faster RCNN Ren et al. (2016) (ResNet50), SSDLite300 (MobileNet v3) |
| **Datasets** | [Training]   COCO v1 Lin et al. (2014) (task),   ImageNet-1k Deng et al. (2009) (backbone) [Evaluation]   Luminus (ImageNet-ES Baek et al. (2024)),   Diverse (*ImageNet-ES Diverse*) | |
| | ResNet50 He et al. (2016), MobileNet v3 Howard et al. (2019), SSDLite300 (Liu et al., 2016; Sandler et al., 2018) | |
| | All models in these experiments were implemented using the pretrained models provided by the Torchvision Marcel & Rodriguez (2010) library. | |

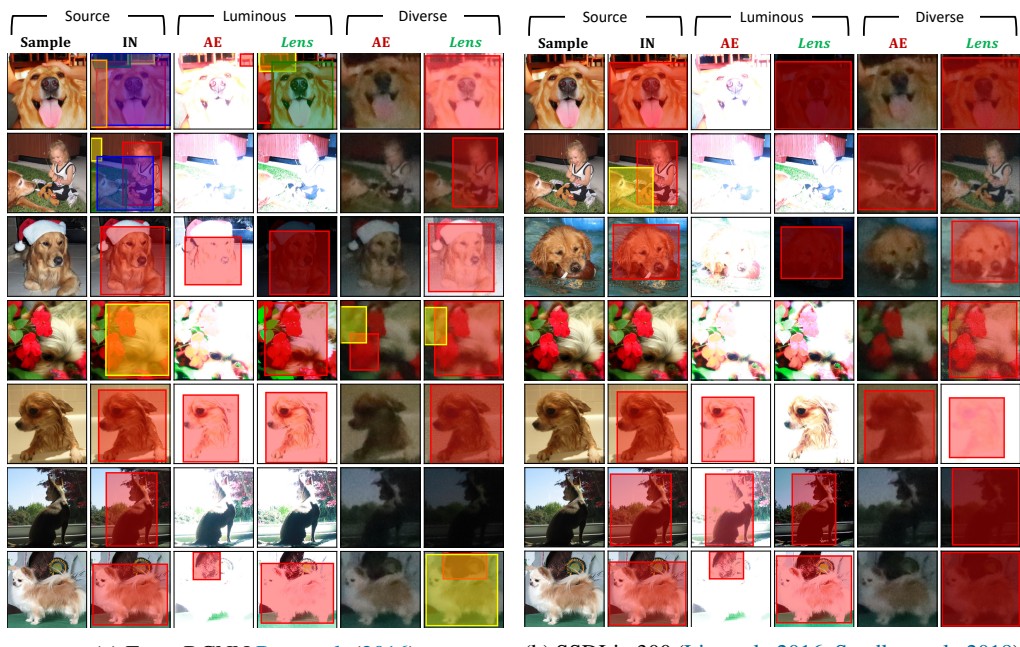

(a) FasterRCNN Ren et al. (2016)  (b) SSDLite300 (Liu et al., 2016; Sandler et al., 2018)

Figure 10: Qualitative Analysis on both Benchmark (Object Detection)

dence is evident in the **performance gap between Oracle-S and *Lens*, suggesting that mitigating overconfidence could further enhance *Lens***. As an initial attempt at sensor control for vision models, *Lens* leverages confidence scores, building on its **generalizability** and **simplicity**. This approach demonstrates substantial potential in two key areas for addressing domain shift problems: real-time applications and ensuring compatibility with diverse camera devices and models. Moving forward, while maintaining the design principles of *Lens*, our research will focus on reducing overconfidence by refining our methodologies and evaluating the approach's adaptability in various real-world environments to improve *Lens*'s reliability and performance. Additionally, as indicated in the ablation study in Section 5.3, existing OOD (Out-of-Distribution) scores address overconfidence stemming from semantic shifts but fail to handle covariate shifts caused by real perturbations (e.g., ImageNet-ES Luminous and Diverse). Therefore, **addressing overconfidence for sensor control requires innovative approaches beyond classical OOD studies**, emphasizing the analysis of intermediate model layers related to low-level features rather than solely focusing on activations in the final layers.

**Addressing Time-constrained Scenarios.** In real-world applications such as autonomous driving and surveillance systems, **rapid environmental shifts** present significant challenges, and **responsiveness** is critical for delivering high-quality service. The responsiveness of *Lens*, which integrates our developed CSA algorithms, depends on the rate of environmental changes. However, by implementing *Lens* within a batch inference system, it can adapt to changes within 0.2 to 0.5 seconds. To achieve more rapid responses, it is necessary to develop CSA algorithms that select a minimal number of options (possibly one or two) with reduced capture times. This represents a promising direction for future research on *Lens*. Successfully adapting sensing systems to time-constrained scenarios requires careful consideration of several additional factors, which can provide potential avenues for future research in this field. In these contexts, it is essential to account for limited available resources and ensure effective scheduling within specified timeframes. This involves balancing trade-offs between accuracy, the number of images captured, and system latency. Furthermore, the latency of each module—such as model inference and image capture—can vary depending on the deployed system architecture and must be meticulously managed to maintain overall system performance. Considering these factors, optimizing CSA algorithms emerges as a promising direction for *Lens*.

### A.2 POTENTIAL OF *Lens* ON NEW FACES

**Addressing Radical Distortion Problems.** In our current study, we did not evaluate radial distortion because it arises independently from light changes caused by environmental factors and sensor control. These factors posed critical issues in real domain shifts, but existing works related to robustness couldn't handle them effectively, making them the primary focus of our investigation. Despite not evaluating radial distortion directly, our methodology has the potential to address it by **controlling framing parameters** such as PTZ (pan, tilt, and zoom). Given two key points, 1) Adjusting pan, tilt, and zoom can minimize radial distortion effects. 2) Our policy algorithm selects the highest-quality images based on camera parameters. Therefore, incorporating framing parameters as control options is expected to effectively manage radial distortion. As a result, jointly applying sensor and framing control could enable the handling of a broader spectrum of domain shifts more effectively. Future research will explore integrating advanced PTZ control algorithms and real-time image quality assessments to further enhance our methodology's robustness against diverse domain shifts.

***Lens* for Representation Learning.** Our method was specifically designed to capture high-quality images in scenarios that utilize model inference results and did not initially consider the high-quality image acquisition processes required for the training stages of representation learning, as suggested in the review. Given that most representation learning pipelines predominantly rely on fine-tuning pre-trained models for downstream tasks, we recognize the possibility of integrating *Lens* during the training stage. This integration could generate customized high-quality images tailored for both pre-trained models and target tasks, potentially reducing data collection costs and enhancing model performance.

## B MORE ANALYSIS

**Label-wise Analysis.** To validate the performance of *Lens* for individual labels in the source domain (ImageNet), we assessed the label-wise accuracy of the target models in Section 5.1 (Experiment 1) for both the representative baseline (AE: Auto Exposure setting) and *Lens*. As illustrated in Figure 11, *Lens* consistently outperforms the baseline, regardless of the performance of individual labels in the source domain. While there are limitations to the improvements when the accuracy in the original sample is excessively low, in most cases, the accuracy enhancements approach those observed in the sampled data (ImageNet Deng et al. (2009): source domain). This pattern is consistent across all datasets and models utilized in our experiments.

**Ablation Study on the Quality Estimator: Confidence (*C*) vs. Entropy of Logits (*E*).** The confidence score and the entropy of logits are interchangeable approaches, as both metrics are based on logits. As shown in Table 5, replacing the *VisiT* score with the entropy of logits yields performance comparable to that of *VisiT* using the confidence score; however, it does not exceed this performance. Therefore, we opted to introduce confidence as a simpler and more representative metric for use in *VisiT* for *Lens*.

## C DETAILS ON *ImageNet-ES Diverse* AND *ES-Studio Diverse* IMPLEMENTATIONS

This section provides details on how *ES-Studio Diverse* is constructed and how *ImageNet-ES Diverse* is collected within the *ES-Studio Diverse* environment. In our previous research (Baek et al., 2024), we developed ES-Studio, enabling individual control over environmental and sensor parameters involved in image acquisition. Utilizing ES-Studio, we compiled ImageNet-ES, a novel dataset comprising 202,000 samples of perturbed data from the environment and camera sensor domains (referred to as "Luminous").

As we mentioned in section 4, to effectively capture the impact of diverse environmental(light) perturbations, we construct *ES-Studio Diverse* and testset of *ImageNet-ES Diverse* (referred to as "Diverse") based on the designs of ES-Studio and ImageNet-ES. We described the common and different settings between Luminous and Diverse on designs for testbed construction and collection configurations in Table 6 and 7.

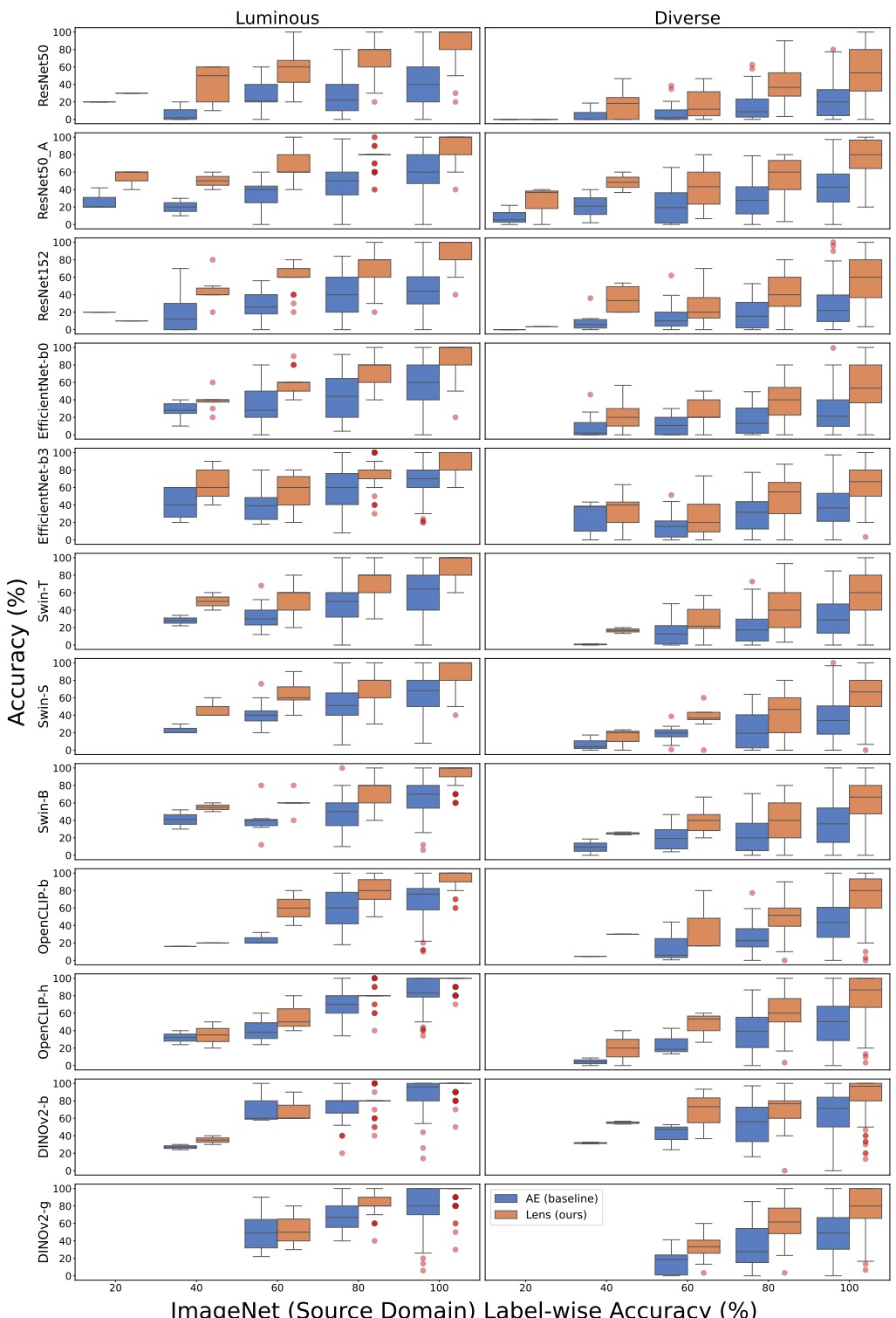

Figure 11: Generalizability of *Lens* Based on Label-wise Performance in the Source Domain.

**The key changes in Diverse** compared to Luminous are: 1) The **display medium** is changed from a **screen to** a 1000 **banners**; and 2) The **lighting options** are expanded from **two to six options**. Detailed changes are described below.

Table 5: Ablation study on *VisiT*: Confidence (***C***) vs. Entropy (***E***) of Logits.

| Model | Num. Params | Pretraining Dataset | DG method | IN | ImageNet-ES Luminous Naive control | | *Lens* | | ImageNet-ES Diverse Naive control | | *Lens* | |
|---|---|---|---|---|---|---|---|---|---|---|---|---|
| | | | | | AE | Random | ***C*** | ***E*** | AE | Random | ***C*** | ***E*** |
| ResNet-50 (He et al., 2016) | 26M | IN-1K | - | 86.0 | 32.1 | 50.4 | **78.6** | **78.8** | 17.6 | 12.0 | **43.3** | **42.9** |
| | | IN-21K | DeepAugment* +AugMix† | 87.0 | 53.2 | 61.4 | **83.1** | **83.2** | 36.2 | 23.6 | **65.1** | **65.2** |
| ResNet-152 (He et al., 2016) | 60M | IN-1K | - | 87.8 | 41.1 | 54.3 | **81.1** | **81.6** | 21.9 | 14.2 | **48.8** | **49.3** |
| EfficientNet-B0 (Tan & Le, 2019b) | 5M | IN-1K | - | 88.2 | 51.8 | 58.3 | **81.2** | **80.5** | 21.8 | 14.0 | **45.9** | **46.3** |
| EfficientNet-B3 (Tan & Le, 2019b) | 12M | IN-1K | - | 88.1 | 62.0 | 66.3 | **83.5** | **82.9** | 33.6 | 21.4 | **55.7** | **55.6** |
| SwinV2-T (Liu et al., 2022) | 28M | IN-1K | - | 90.6 | 54.1 | 63.1 | **82.6** | **82.1** | 26.5 | 16.9 | **50.3** | **50.0** |
| SwinV2-S (Liu et al., 2022) | 50M | IN-1K | - | 91.7 | 59.9 | 65.5 | **84.5** | **84.9** | 30.8 | 18.9 | **55.6** | **55.8** |
| SwinV2-B (Liu et al., 2022) | 88M | IN-1K | - | 91.9 | 60.0 | 65.5 | **85.3** | **85.1** | 30.8 | 18.5 | **55.3** | **55.3** |
| OpenCLIP-b (Cherti et al., 2023) | 87M | LAION-2B | Text-guided pretrain | 94.3 | 66.1 | 71.3 | **90.9** | **90.4** | 38.8 | 24.5 | **67.6** | **67.1** |
| OpenCLIP-h (Cherti et al., 2023) | 632M | LAION-2B | | 94.9 | 79.0 | 77.6 | **93.0** | **92.9** | 45.5 | 29.3 | **74.4** | **74.5** |
| DINOv2-b (Oquab et al., 2023) | 90M | LVD-142M | Dataset curation | 93.6 | 74.5 | 73.9 | **90.6** | **91.1** | 44.5 | 28.3 | **72.9** | **73.9** |
| DINOv2-g (Oquab et al., 2023) | 1.1B | LVD-142M | | 94.7 | 84.3 | 79.8 | **93.1** | **93.4** | 62.8 | 35.3 | **82.9** | **83.5** |
| All models | | | | 90.7 | 59.8 | 65.6 | **85.6** | **85.5** | 34.2 | 21.4 | **59.8** | **59.9** |

Luminous: Baek et al. (2024), *:(Hendrycks et al., 2021), †: (Hendrycks et al., 2019), IN: ImageNet (Le & Yang, 2015), AE: Auto exposure

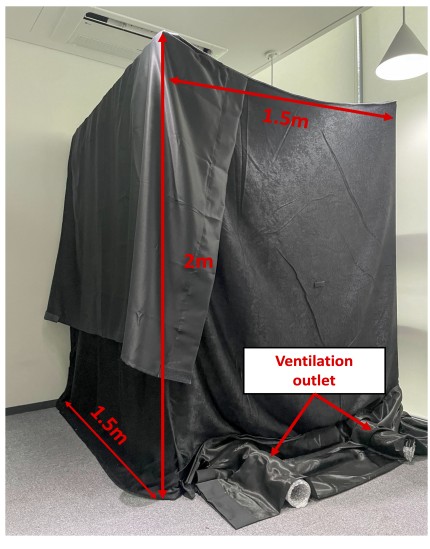

(a) External.

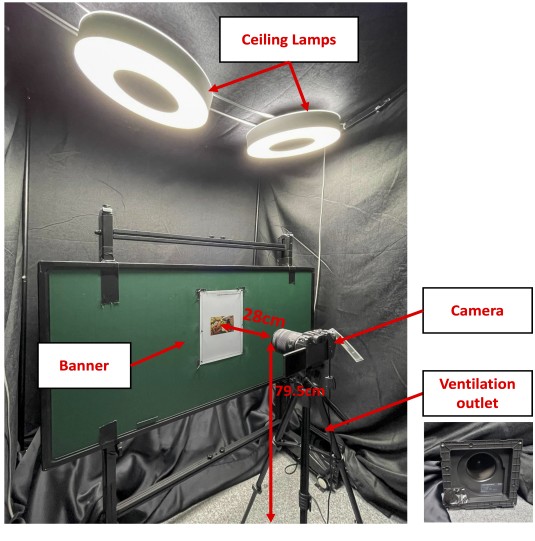

(b) Internal.

Figure 12: Actual appearance of *ES-Studio Diverse*.

- **Banner:**
  - **Print & Material**: As shown in Figure 13a, each sample image from the Tiny-ImageNet (Le & Yang, 2015) subset was printed on a separate banner with A4-size (210 mm x 297 mm) with 600 DPI, precisely centered while maintaining the original image's aspect ratio. A total of 1,000 banners were produced, each containing a single sample image. To prevent image quality degradation due to the printing process and paper material properties, we carefully tested various DPIs (72, 300, 600) and paper types. We ultimately selected 600 DPI because it provided the best image quality and minimized distortions and noise caused by reduced

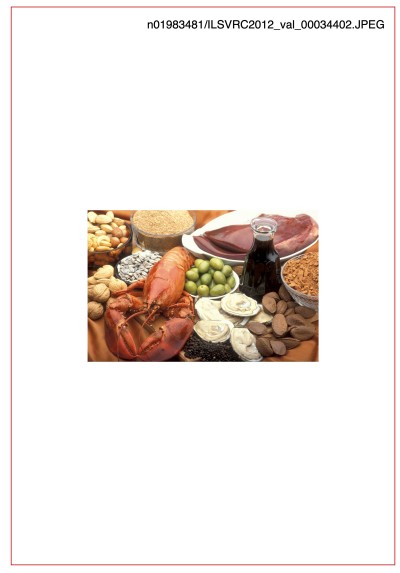

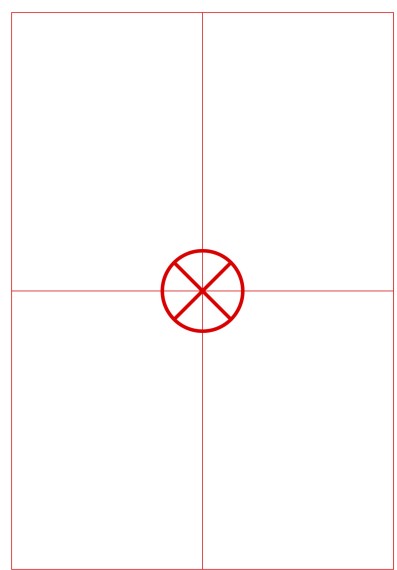

(a) Sample banner.  (b) Alignment banner.

Figure 13: Banner examples of *ImageNet-ES Diverse*.

Table 6: Comparison on Testbed between Luminous and Diverse

| Comparison | Components | ES-Studio | *ES-Studio Diverse* | Specifications |
|---|---|---|---|---|
| Different | Display for reference (Alteration) | Screen (by controller) | Banner (by humans) | Screen: 55-inch OLED 4K UHD TV (LG OLED55B3FNA) Banner: 1000 A4-sized PVC banners with 600 DPI (1 banner/sample) |
| Common | Darkroom Environment factor Sensor factor Control system | Completely enclosed dark w/ blackout fabric Brightness controllable ceiling lamps Sensor controllable camera Desktop Computer + Wifi Network | | 1.5 m × 1.5 m × 2 m Philips Hue White & Color Ambiance Infuse (Two lamps) 'Canon EOS-RP' body + 'RF 24-105mm F4-7.1 IS STM' lens Apple Mac Studio M2 Max + CCAPI* + Philips Hue API |

∗CCAPI: Canon camera control API

expressiveness during printing, among the three DPIs tested. We also chose a PVC banner, as it is resistant to light reflection, humidity, and creasing.

- **Placement:** In a darkroom of *ES-Studio Diverse*, the banner is fixed in place (Figure 12). To prevent image distortion, the camera's height is carefully adjusted, ensuring it's positioned 28 cm away from the banner in a straight line. Furthermore, the banner is securely attached to a fixed position on a magnetic blackboard using six magnets. To minimize framing-induced distortion, we use an "Alignment banner (Figure 13b)" to adjust the camera angle, keeping the camera grid and the banner's support line parallel.

- **Cropping Process:** Aside from the cropping process, the preprocessing steps are identical to those in Luminous preprocessing Baek et al. (2024), labeling the images using the data collection log. The cropping process extracts the valid area from the collected images using the ROI coordinates. Since the banner and camera framing are fixed, the ROI is identical for captured images with the same reference banner. We developed and use an interactive tool to obtain the ROI coordinates: 1) Zoom in on a captured image (taken with AE) for each sample. 2) Using a drag interaction, select the valid area, ensuring it includes the sample but has minimal surrounding area. 3) Crop the images for all settings (6 environments, 27 parameter options, and 5 AE shots) using the ROI coordinates.

• **Lighting Options:** As shown in Table 4c, five additional lighting options, varying in magnitude and direction, are defined. L5 (Turn off, included in Luminous) option is excluded from Diverse because it produces blacked-out, uninformative images in non-luminous scenes.

Except for these two aspects, all other components and configuration settings are the same as in ImageNet-ES. Moreover, the validation process is the same as that used for ImageNet-ES, and Diverse was validated by five individuals to assess the integrity of the collected data. Details for the common

Table 7: Comparison on Configurations of testset collection between Luminous and Diverse

| Comparison | Collection Configurations | ImageNet-ES | *ImageNet-ES Diverse* |
|---|---|---|---|
| Different | Light Options | L1 (on) & L5 (off) | L1-L7 w/o L5 |
| | Caputured images | 64,000 | 19,2000 |
| Common | Reference Samples | 1000 samples (Randomly selected 5 samples/class in TinyImageNet validation set) | |
| | Frame Setting | 6240 × 4160 pixels, AF (Auto focus) mode (w/ metering mode) | |
| | Sensor Options | AE: 5 Shots, M: 27 parameter options (controlled parameters: Shutter speed, Aperture, ISO) | |

∗AE: Auto Exposure, M: Manual Control (Details in Table 8)

Table 8: Manual camera sensor parameter setting of test set of Luminous and Diverse

| No. | 1 | 2 | 3 | 4 | 5 | 6 | 7 | 8 | 9 | 10 | 11 | 12 | 13 | 14 | 15 | 16 | 17 | 18 | 19 | 20 | 21 | 22 | 23 | 24 | 25 | 26 | 27 |
|---|---|---|---|---|---|---|---|---|---|---|---|---|---|---|---|---|---|---|---|---|---|---|---|---|---|---|---|
| ISO | 250 | 2K | 16K | 250 | 2K | 16K | 250 | 2K | 16K | 250 | 2K | 16K | 250 | 2K | 16K | 250 | 2K | 16K | 250 | 2K | 16K | 250 | 2K | 16K | 250 | 2K | 16K |
| SS | 1/4″ | 1/4″ | 1/4″ | 1/60″ | 1/60″ | 1/60″ | 1/1K″ | 1/1K″ | 1/1K″ | 1/4″ | 1/4″ | 1/4″ | 1/60″ | 1/60″ | 1/60″ | 1/1K″ | 1/1K″ | 1/1K″ | 1/4″ | 1/4″ | 1/4″ | 1/60″ | 1/60″ | 1/60″ | 1/1K″ | 1/1K″ | 1/1K″ |
| A | f5.0 | f5.0 | f5.0 | f5.0 | f5.0 | f5.0 | f5.0 | f5.0 | f5.0 | f9.0 | f9.0 | f9.0 | f9.0 | f9.0 | f9.0 | f9.0 | f9.0 | f9.0 | f16 | f16 | f16 | f16 | f16 | f16 | f16 | f16 | f16 |

∗SS: Shutter speed, A: Aperture

design aspects of both testbeds and datasets are described in the supplementary material of our previous research (Baek et al., 2024).

# D    DETAILS ON SCENE SPECIFIC CAMERA CONTROL CONCEPTS

This section details the Model-Scene specific camera control concepts within our framework. As illustrated in Figure 14, the design of *VisiT* is scene-specific, acknowledging that optimal parameters can vary significantly across different scenes, even when utilizing the same underlying model. The core principle of scene-specific sensor control is that a universally optimal, fixed parameter set does not exist. Our solution space analysis (Figures 8g and 8h) demonstrably shows that optimal parameters differ drastically between, for instance, the "Luminous" and "Diverse" scenes. This highlights the necessity of dynamically adapting control strategies to the specific characteristics of each scene.

# E    DETAILS ON EXPERIMENTAL SETUPS AND RESULTS

## E.1    ANOTHER BENCHMARK: IMAGENET-ES

Table 9: Environment and Sensor specifics of ImageNet-ES (Baek et al., 2024).

| Dataset | Original samples | Light | Camera sensor | ISO | Shutter speed | Aperture | Captured images |
|---|---|---|---|---|---|---|---|
| Test | 1,000 (5 samples/class) | On/Off | Auto exposure (5 shots) Manual (27 options) | Auto 250/2000/16000 | Auto (1/4″)/(1/60″)/(1/1000″) | Auto f5.0/f9.0/f16 | 10,000 54,000 |
| Validation | 1,000 (5 samples/class) | On/Off | Auto exposure (5 shots) Manual (64 options) | Auto 200/800/3200/12800 | Auto (0″4)(1/20″)/(1/160″)/(1/1250″) | Auto f5.0/f8.0/f13/f20 | 10,000 128,000 |

In this paper, two test datasets were used, both of which include extensive natural perturbations in environmental and sensor domains, incorporating 27 manual controls and 5 auto-exposure shots.

For **ImageNet-ES** (Baek et al., 2024), a subset of Tiny-ImageNet (Le & Yang, 2015) was displayed on a monitor and captured using a camera. During the process, lighting conditions were varied by turning the lights on and off, and camera parameters were adjusted. Details of the environment and sensor specifications are provided in Table 9. Five images were randomly selected from each of the 200 classes in the Tiny-ImageNet validation set (Le & Yang, 2015). To ensure visual fidelity, each sampled image had a resolution greater than 375 × 500 pixels, avoiding distortion when displayed on the TV screen. In total, 1,000 samples were collected for the test dataset and these samples are used identically in constructing the test set of *ImageNet-ES Diverse* . For the validation set, we employed the same process on a non-overlapping set of 10,000 samples, utilizing 64 distinct parameter options to create the ImageNet-ES validation set. Further details can be found in our previous research (Baek et al., 2024). As a potential future work, the *ImageNet-ES Diverse* validation set could be constructed by applying the six light options of *ImageNet-ES Diverse* to the same 10,000 samples and 64 parameter options used in the ImageNet-ES validation set.

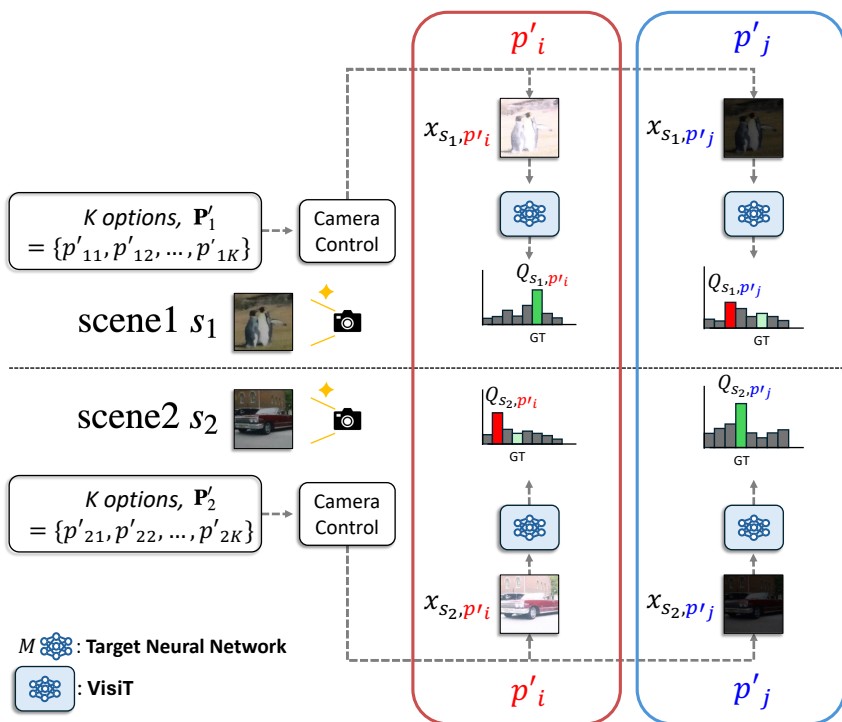

Figure 14: Scene-specific design of *VisiT*.

## E.2 EXPERIMENT SETUPS OF OUT-OF-DISTRIBUTION (OOD) DETECTION.

### E.2.1 OOD (OUT-OF-DISTRIBUTION) DETECTION FRAMEWORK SETTINGS AND TECHNIQUES

**OOD Detection Framework and Datasets.** To leverage OOD detection techniques as a proxy, we employed the Semantic-Centric framework (Yang et al., 2022), a prevalent and widely-adopted approach in OOD research. Existing studies predominantly focus on classifying samples as either belonging to classes encountered during training or not. Consequently, we structured the dataset configuration for OOD detection training and validation as detailed in Table 10. For evaluating the efficacy of OOD detection methods as proxies for validation (quality indicator) and testing (ablation on *VisiT*), we partitioned the Tiny-ImageNet (Le & Yang, 2015) (ImageNet-ES reference dataset) validation set into three distinct subsets: S1, S2, and S3. The validation and test splits of ImageNet-ES were assigned the same images as S1 and S2, respectively. The remaining images, comprising 40 images per class, were designated as S3. Given that Tiny-ImageNet images are provided in a resized 64x64 format, corresponding images from the original ImageNet dataset were utilized to maintain the native resolution. This partitioning strategy for Tiny-ImageNet is further described in Table 11.

**OOD Detection methods.** We validate OOD detection techniques on validation set of ImageNet-ES, including ViM (Wang et al., 2022a), ReAct (Sun et al., 2021), ASH (Djurisic et al., 2023) and MSP (Hendrycks & Gimpel, 2017). These methods demonstrate state-of-the-art performance and serve as baselines in recent OOD research. To validate current OOD detection methods, we leverage the results and APIs provided by OpenOOD (Yang et al., 2022). All implementations are based on the OpenOOD package.

### E.2.2 MODELS

We selected four models with diverse architectures, widely adopted in OOD detection research, as our underlying models to adapt OOD detection methods. Details of these models are provided in Table 12. All model weights used in the OOD detection experiments were obtained from the timm library (Wightman, 2019). As the pre-trained weights produce predictions for 1,000 classes, we fine-tuned the classifier of each model to output 200 classes, corresponding to Tiny-ImageNet. Non-resized images from the Tiny-ImageNet training set were used for fine-tuning, with the feature

Table 10: Datasets Used in OOD Detection Experiments

| Experiment Setting | | Train | | Validation | | Test |
| | ID | OOD | ID | OOD | Quality Estimator | Ablation Study for *VisiT* |
|---|---|---|---|---|---|---|
| Semantics centric | S3 | OpenImage-O (train) | S1 | Textures (test) | $val_{ImageNet-ES}$ | $test_{ImageNet-ES}$ |

Table 11: Description of partitions of Tiny-ImageNet validation set (10K samples)

| Partition | S1 Ref.$val_{\text{ImageNet-ES}}$ | S2 Ref.$test_{\text{ImageNet-ES}}$ | S3 $(val_{\text{Tiny-ImageNet}} \backslash (S1 \cup S2))$ |
|---|---|---|---|
| # of samples | 1,000 | 1,000 | 8,000 |

Table 12: Description of underlying models for OOD detection experiments. (Optimizer: SGD, Scheduler: ReduceLROnPlateau, Batch size: 128)

| Model | # of params | Pretrained | Acc. on $Val_{Tin}$ | Training configuration |
|---|---|---|---|---|
| EfficientNet-B0 Tan & Le (2019a) | 4.3M | | 86.2% | lr: $5 \times 10^{-3}$, epochs: 20 |
| ResNet18 He et al. (2016) | 11.3M | ImageNet-1K | 82.4% | lr: $5 \times 10^{-2}$, epochs: 15 |
| DeiT Touvron et al. (2020) | 86M | | 91.2% | lr: $5 \times 10^{-3}$ , epochs: 20 |
| Swin-B Liu et al. (2021a) | 86.9M | | 94.2% | lr: $5 \times 10^{-3}$ , epochs: 20 |

extractor of each model frozen during this process. The specific training configuration and final accuracy are also presented in Table 12.

### E.2.3 FURTHER ANALYSIS

Figure 15 illustrates the correlation between proxy candidates and image quality on the light Vision Transformer model (DeiT Touvron et al. (2020)), following the methodology outlined in Section 3.2. Consistent with the findings presented in Figure 3, a higher confidence score exhibits a strong positive correlation with classification accuracy.

### E.3 EXPERIMENT SETUPS OF GENERALIZABILITY

### E.3.1 TARGET MODELS SETUP

For all generalization experiments, model weights were sourced from PyTorch. Exceptions to this were OpenCLIP-b/h and the DG variant (DeepAugment Hendrycks et al. (2021) + Augmix Hendrycks et al. (2019)) of ResNet-50, which were obtained from timm Wightman (2019) and Hendrycks et al. (2019), respectively.

### E.4 EXPERIMENT SETUPS OF REAL-TIME ADAPTATIONS

### E.4.1 TEST-TIME ADAPTATION (TTA) TECHNIQUES

In this section, we describe details of the test-time adaptation techniques employed in the real-time adaptation experiments in section 5.2.

**Prediction-time Batch Normalization (BN1)**(Nado et al., 2020): Improves model robustness to covariate shifts by updating Batch Normalization statistics during prediction. It is computationally efficient as it does not require backward propagation.

**Batch Normalization Adaptation (BN2)** (Schneider et al., 2020): Dynamically updates the running statistics (running mean and variance) of Batch Normalization during inference. Instead of using test batch statistics in isolation, it continuously updates the running mean and variance as test data is encountered, making adaptation more flexible under varying test batches.

**Fully Test-time Entropy Minimization (Tent)** (Wang et al., 2021): Adapts to domain shifts during test time by minimizing the output entropy of predictions. It updates not only the BN statistics but also the entire model's parameters, providing greater robustness across a broader range of test samples, but increasing computational costs.

### E.4.2 MODELS

Of the models used in Section 3, we selected two representative lightweight architectures, ResNet-18 (He et al., 2016) and EfficientNet-B0 (Tan & Le, 2019b), for this experiment. These models were chosen due to their compatibility with our three TTA methods, as they include Batch Normalization (BN) layers required for the application of these techniques.

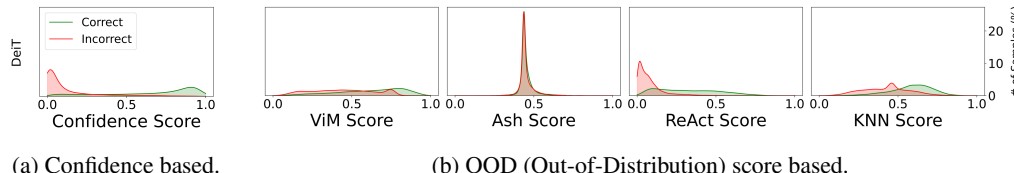

(a) Confidence based.        (b) OOD (Out-of-Distribution) score based.

Figure 15: Proxy for image quality assessment on DeiT: Each score is normalized between 0 to 1.

Table 13: Detailed results of real-time adaptation for ResNet18 He et al. (2016)

| Env. | k | 1 | 2 | 3 | 4 | 5 | 6 | 7 | 8 | 9 | 10 | 11 | 12 | 13 | 14 | 15 | 16 | 17 | 18 | 19 | 20 | 21 | 22 | 23 | 24 | 25 | 26 | 27 |
|---|---|---|---|---|---|---|---|---|---|---|---|---|---|---|---|---|---|---|---|---|---|---|---|---|---|---|---|---|
| | CSA1 | 46.8 | 61.8 | 68.7 | 70.2 | 71.9 | 73.4 | 73.3 | 73.1 | 73.2 | 73.6 | 74.1 | 73.6 | 73.8 | 74.0 | 73.9 | 73.9 | 73.2 | 74.2 | 73.9 | 73.9 | 73.9 | 73.8 | 74.1 | 73.7 | 73.9 | 73.8 | 73.8 |
| Luminous | CSA2 | 47.0 | 61.8 | 68.3 | 70.9 | 72.2 | 72.6 | 72.6 | 72.9 | 72.6 | 73.1 | 72.9 | 73.2 | 73.1 | 73.6 | 73.6 | 73.4 | 73.4 | 73.6 | 73.7 | 73.9 | 73.9 | 73.5 | 73.8 | 73.9 | 73.6 | 73.7 | 73.8 |
| | CSA3 | 42.5 | 60.1 | 67.2 | 69.9 | 71.0 | 71.2 | 72.4 | 72.3 | 72.6 | 73.1 | 73.1 | 73.2 | 73.2 | 73.3 | 73.4 | 73.5 | 73.8 | 73.7 | 73.9 | 73.8 | 73.7 | 73.8 | 73.6 | 73.7 | 73.8 | 73.8 | 73.8 |
| | CSA1 | 9.3 | 15.3 | 19.7 | 22.7 | 24.9 | 26.9 | 28.1 | 29.8 | 30.0 | 31.0 | 31.2 | 31.6 | 32.4 | 32.5 | 33.2 | 33.1 | 33.2 | 33.3 | 33.7 | 33.7 | 34.0 | 34.4 | 34.0 | 34.5 | 34.3 | 34.4 | 34.6 |
| Diverse | CSA2 | 8.8 | 15.3 | 19.5 | 23.2 | 25.2 | 27.4 | 28.8 | 25.9 | 26.9 | 28.0 | 28.3 | 28.9 | 29.1 | 29.7 | 30.2 | 30.6 | 31.1 | 31.3 | 32.1 | 33.7 | 31.8 | 32.2 | 32.9 | 33.0 | 34.0 | 34.1 | 34.6 |
| | CSA3 | 0.6 | 1.1 | 1.4 | 1.8 | 1.9 | 2.1 | 2.5 | 2.6 | 2.9 | 8.4 | 12.4 | 16.4 | 17.7 | 20.1 | 22.1 | 23.3 | 24.2 | 25.1 | 27.9 | 29.2 | 30.6 | 31.9 | 33.1 | 33.3 | 34.0 | 34.3 | 34.6 |

## E.5 DETAILED RESULTS FOR EXPERIMENTS ON REAL-TIME ADAPTATIONS.

Tables 13 and 14 detail the performance of real-time adaptation results for the *Lens* system as the number of input images, k, varies across the complete range from 1 to 27.

Table 14: Detailed results of real-time adaptation for EfficientNet-B Tan & Le (2019b)

| Env. | k | 1 | 2 | 3 | 4 | 5 | 6 | 7 | 8 | 9 | 10 | 11 | 12 | 13 | 14 | 15 | 16 | 17 | 18 | 19 | 20 | 21 | 22 | 23 | 24 | 25 | 26 | 27 |
|---|---|---|---|---|---|---|---|---|---|---|---|---|---|---|---|---|---|---|---|---|---|---|---|---|---|---|---|---|
| | CSA1 | 49.8 | 68.4 | 73.8 | 77.0 | 76.6 | 76.2 | 77.2 | 78.3 | 77.2 | 77.9 | 78.2 | 77.2 | 78.2 | 77.8 | 78.4 | 77.6 | 77.8 | 77.5 | 78.1 | 78.1 | 77.9 | 77.8 | 77.6 | 77.8 | 77.8 | 77.8 | 77.8 |
| Luminous | CSA2 | 50.0 | 67.2 | 72.8 | 76.1 | 76.2 | 77.4 | 77.5 | 78.3 | 77.2 | 78.1 | 77.9 | 77.4 | 77.6 | 77.7 | 77.6 | 77.5 | 78.1 | 77.9 | 77.7 | 78.1 | 77.7 | 77.6 | 77.5 | 77.6 | 77.8 | 77.8 | 77.8 |
| | CSA3 | 47.3 | 66.1 | 72.7 | 75.9 | 76.4 | 77.0 | 77.5 | 77.5 | 77.6 | 77.8 | 78.6 | 78.4 | 78.6 | 78.4 | 78.8 | 78.6 | 78.6 | 78.6 | 78.4 | 78.3 | 78.5 | 77.9 | 78.1 | 77.9 | 77.8 | 77.8 | 77.8 |
| | CSA1 | 11.3 | 19.7 | 24.5 | 28.4 | 30.8 | 32.4 | 33.9 | 34.9 | 36.0 | 36.2 | 36.4 | 37.5 | 37.6 | 38.4 | 38.4 | 38.6 | 38.8 | 39.2 | 39.1 | 39.2 | 39.6 | 39.2 | 39.2 | 39.2 | 39.5 | 39.8 | 39.6 |
| Diverse | CSA2 | 11.2 | 19.4 | 24.8 | 28.7 | 30.7 | 32.9 | 33.2 | 31.1 | 32.0 | 33.1 | 33.5 | 34.3 | 35.0 | 35.3 | 35.8 | 36.1 | 36.2 | 36.5 | 37.0 | 39.2 | 37.8 | 38.0 | 37.8 | 38.7 | 38.8 | 39.2 | 39.6 |
| | CSA3 | 1.1 | 1.6 | 2.2 | 2.8 | 3.3 | 3.6 | 4.0 | 4.5 | 4.9 | 11.7 | 16.7 | 20.7 | 23.5 | 26.0 | 27.6 | 29.2 | 30.3 | 31.4 | 33.7 | 35.6 | 36.5 | 37.1 | 38.1 | 38.8 | 39.0 | 39.4 | 39.6 |

## F ADDITIONAL REPRESENTATIVE EXAMPLES FROM *ImageNet-ES Diverse*.

More *ImageNet-ES Diverse* examples are provided in Figure 16.

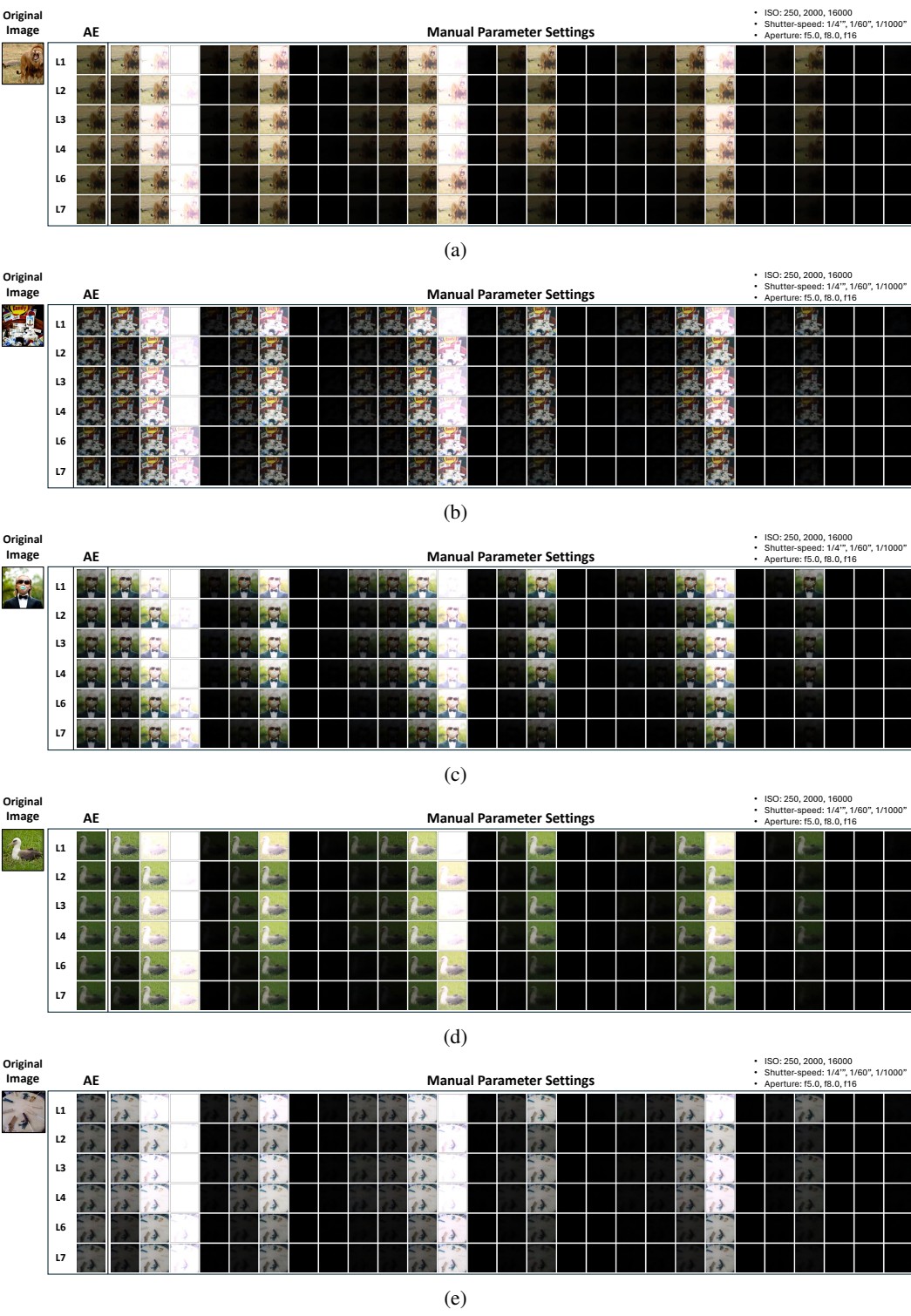

Figure 16: *ImageNet-ES Diverse* samples (a)-(e): Each subfigure demonstrates how image characteristics (e.g., brightness, color, sharpness) change based on the sensor parameter settings. Even in manual mode, the image varies significantly depending on environmental conditions (L1-L7, excluding L5) and parameter adjustments. Although auto exposure fails to produce high-quality images for neural networks, it provides images of consistent quality for humans across various environments. This indicates that sensor parameters and environmental conditions significantly influence the image quality for both neural networks and humans.

