# OpenReview forum: "Adaptive Camera Sensor for Vision Models"
_ICLR.cc/2025/Conference — ICLR 2025 Poster_

### Official Review · Reviewer_5LQ1 · 2024-10-24

**Soundness:** 3
**Presentation:** 3
**Contribution:** 3
**Rating:** 6
**Confidence:** 4

**Summary:**

This paper proposes a camera sensor control algorithm that actively collects high-quality images based on the model’s perspective. The authors build a training-free, model-specific quality indicator called VisiT to evaluate unlabeled samples at test time without adaptation cost. They also propose a new benchmark based on ImageNet-ES that captures the natural perturbations from changes of sensor and lighting conditions. The experimental results show the effectiveness of the proposed method.

**Strengths:**

+ This work provides a new perspective for TTA by improving the sensor parameters, which is very interesting to strengthen the performance of foundation models.

+ The proposed benchmark provides a fair and diverse comparison for the generalization ability of the TTA in more scenarios.

+ The experimental results show the TTA can increase the accuracy by a sizable margin with different vision models.

**Weaknesses:**

- The novelty of this paper is limited. Leveraging the logits as a confidence score is widely adopted in many uncertainty-based methods for many applications such as classification and detection. Besides that, I don’t see other technical contribution from the proposed method.

- More details about the used techniques should be given for readers to better understand the intuition of this method. For example, how do you generate the camera parameter candidates? And how do you guarantee the discrete candidate can cover the optimal solution in the continuous parameter space?

- As evaluating the confidence score requires one forward pass of the image, deploying the Lens method seems to significantly increase the inference latency of the original model. Although subset selection method is used in CSA, it may miss the optimal solution. How do you deal with the trade-off?

- Some visualization could be added to show the best camera parameters of different input images and different backbones. Analysis to the results of selection can also be added to give readers more intuition.

- Have you ever tried other proxies that are often used in uncertainty learning such as entropy of logits?

**Questions:**

See weakness

---

> ### Author Response · Authors · 2024-11-23
> **Response to Technical Contribution**
>
> We sincerely appreciate your time and effort in providing us with positive comments. We respond to your question in what follows. We also ask you to kindly refer to the common response we have posted together.
> ***
> **\[Weakness 1]** The novelty of this paper is limited. Leveraging the logits as a confidence score is widely adopted in many uncertainty-based methods for many applications such as classification and detection. Besides that, I don’t see other **technical contribution** from the proposed method.
>
> **\[Response]**
>
> While leveraging logits as a confidence score is common in many uncertainty-based methods, our approach distinguishes itself in several key ways:
>
> **Addressing Domain Shift with Sensor Control**: Our method is **the first** to effectively tackle the domain shift problem in real-world scenarios **by introducing model- and scene-specific sensor control**. Unlike existing robustness techniques that focus solely on model training to handle domain shifts, our holistic approach integrates sensor adjustments tailored to both the model and the scene. This method overcomes the limitations of traditional approaches that do not account for environmental and sensor variations, **enhancing classification accuracy by up to 51.31%.** Additionally, our work demonstrates the ability to **compensate for up to 50× difference in model size** through real-time sensor control, **underscoring the significance of the proposed technical concepts**.
>
> **Generalizable and Simple Sensor Control Strategy**: To the best of our knowledge, **no prior work has established criteria or methodologies** specifically designed for sensor control in vision models to address domain shifts. Our sensor control strategy is both **generalizable** across different model architectures and camera devices and **simple** to adopt. We introduce a **novel framework** that dynamically adjusts camera sensor parameters **based on real-time scene analysis**, leading to significant performance improvements independent of the model architecture. This constitutes a clear technical advancement over existing methods.
>
> **System Design Considered Real-Time Adaptation**: We have **thoroughly analyzed the potential for real-time adoption** of our methodology. Our system, Lens, adapts to real-world conditions through sophisticated system designs, such as the **Camera System Adaptation (CSA)** module detailed in Section 5.2. The CSA module enables seamless integration of sensor control with model inference by dynamically adjusting camera parameters in real-time based on environmental feedback. This ensures that our system **maintains high performance even in dynamic environments while keeping operational costs low**. This capability represents a significant technical contribution, **providing a practical solution** for deploying robust vision models in real-world applications.
>
> **In summary**:
>
> Our work offers novel contributions by combining sensor control with model-based confidence scoring to address domain shifts, introducing a unique and generalizable sensor control strategy, and ensuring real-time adaptability through advanced system design. These elements collectively enhance the robustness and applicability of vision models in diverse and evolving environments.
>
> We have enhanced these aspects in Section 1 and Appendix A.1.2 of the revised version.

---

> ### Author Response · Authors · 2024-11-23
> **Response to Details for Techniques**
>
> **\[Weakness 2]** More details about the used techniques should be given for readers to better understand the intuition of this method. For example, how do you generate the camera parameter candidates? And how do you guarantee the discrete candidate can cover the optimal solution in the continuous parameter space?
>
> **\[Response]**
>
> In Section 3.1, we provide a comprehensive overview of the Lens pipeline, detailing how the Camera System Adaptation (CSA) framework is integrated into Lens. Specifically, **in Section 5.2, we introduce three distinct CSAs** designed to generate camera parameter candidates **independently of the parameter space**. Our experiments demonstrate that our strategy allows Lens to achieve a sub-optimal yet highly effective balance between accuracy and computational efficiency in real-time adaptation scenarios.
>
>  We acknowledge that exploring methods to more precisely cover the optimal solution in the continuous parameter space is an important area for future research. However, factors such as system latency and adaptability, including training and inference times, which are the advantages of the current version of Lens, need to be jointly considered to develop practical solutions.
>
> We have enhanced these aspects in revised appendix A.1.1.

---

> ### Author Response · Authors · 2024-11-23
> **Addressing Trade-off from CSAs**
>
> **\[Weakness 3]** As evaluating the confidence score requires one forward pass of the image, deploying the Lens method seems to significantly increase the inference latency of the original model. Although subset selection method is used in CSA, it may miss the optimal solution. How do you deal with the trade-off?
>
> **\[Response]**
>
> **Our system is specifically designed for batch inference**, which ensures that integrating the Lens method **does not add any extra inference costs** beyond the existing processing pipeline. Thus, in our cost analysis, we focused on the latency associated with capturing and processing images, carefully evaluating both the number of photos taken and the time required for image acquisition.
>
> Regarding the subset selection method used in CSA, we recognize that limiting the parameter exploration can lead to suboptimal solutions. However, these approach allows us to **balance accuracy and latency effectively**. As demonstrated in Section 5.2, these CSA algorithms maintain high accuracy while significantly reducing latency by narrowing down the parameter search space. **This trade-off enables real-time performance without sacrificing the reliability of the results.**
>
> We have further enhanced the explanation of these points in Section 3.1 of the revised paper to clarify how we manage the balance between accuracy and efficiency.

---

> ### Author Response · Authors · 2024-11-23
> **Analysis for Model- and Scene- Specific Solution Space**
>
> **\[Weakness 4]** Some visualization could be added to show the best camera parameters of different input images and different backbones. Analysis to the results of selection can also be added to give readers more intuition.
>
> **\[Response]**
>
> **In Section 5.4 (Figure 8), we show that model- and scene-specific parameter control is crucial**. We demonstrated this by varying factors like input images and backbone architectures and analyzing the VisiT scores for each parameter option in the solution space. We have enhanced the explanation of these points in the final manuscript.

---

> ### Author Response · Authors · 2024-11-23
> **Using Entropy of Logits as Proxy**
>
> **\[Weakness 5]** Have you ever tried other proxies that are often used in uncertainty learning such as entropy of logits?
>
> **\[Response]**
>
> **The confidence score and the entropy of logits are interchangeable approaches**, as both metrics are based on logits. In our assessment of target model performance in experiments on the generalizability of Lens, we found that replacing the VisiT score with the entropy of logits provides **comparable performance**. We chose to introduce confidence as a simpler and more representative metric for use in VisiT for Lens. In response to your feedback, **we've incorporated the results in the Appendix B** to include an ablation study of VisiT, which demonstrates the performance when replacing the confidence score with the entropy of logits.

---

> ### Author Response · Authors · 2024-11-28
> **Kind Reminder**
>
> Dear reviewer 5LQ1,
>
> As the PDF upload deadline has passed and we have uploaded the latest revised manuscript, we kindly remind you that we are awaiting your response. We hope that your concerns have been addressed by our rebuttal and revised manuscript, and welcome further discussion during the remaining period. We sincerely appreciate your efforts and thank you again for your time and consideration.
>
> Best regards,
>
> Authors.

---

> > ### Author Response · Authors · 2024-12-03
> > **Last Reminder**
> >
> > Dear reviewer 5LQ1,
> >
> > We believe that our rebuttal has addressed your concerns.
> >
> > As the discussion deadline is approaching, please put your response as soon as possible.
> >
> > We appreciate your efforts.
> >
> > Best regards,
> >
> > Authors

---

### Official Review · Reviewer_v7Km · 2024-11-03

**Soundness:** 3
**Presentation:** 4
**Contribution:** 3
**Rating:** 5
**Confidence:** 2

**Summary:**

The authors propose  Lens, a camera sensor control method that enhances model performance by capturing
high-quality images from the model’s perspective, rather than relying on traditional human-centric sensor control.

Thought is well received.

Lens is lightweight and adapts sensor parameters to specific models and scenes in real-time training-free, Lens uses a model-specific quality indicator (VisiT) that evaluates individual unlabeled samples at test time using confidence scores, without additional adaptation costs.
A new benchmark ImageNet-ES Diverse is introduced to validate Lens.

Lens is claimed to be a novel adaptive sensor control system that captures high-quality images robust to real-world perturbations.
The core idea of Lens is to identify optimal sensor parameters that allow the target neural network to better discriminate between
objects, akin to adjusting a pair of glasses for clear vision.

Detailed experiments have been done, including real physical time analysis.


The limitations are well received.

I just have one thought, would your method be robust against perspective and radial distortion both.

**Strengths:**

Paper is well written and easy to follow and understand .
Detailed experiments are there.
Two datasets have been investigated including one that was introduced by the authors.

**Weaknesses:**

I do not see much weakness. I am simply thinking about radial distortion.
Another thought is how easy it is to incorporate the proposed methods in SSL, multi, and Meta-learning.
Perspective distortion is evident in real-world applications, as indicated in the following papers
a) Möbius Transform for Mitigating Perspective Distortions in Representation Learning
b) LCM: Log Conformal Maps for Robust Representation Learning to Mitigate Perspective Distortion
Radial distortion is often evident as well when using cameras like fisheye view.
Both distortions distort the visual concepts, and therefore, the models do not perform well.

Authors are not forced to do experiments. However, they are encouraged to comment on such situations and whether 2'Lens' would be robust against them as well.

**Questions:**

Please follow weaknesses and summary.

---

> ### Author Response · Authors · 2024-11-23
> **Addressing Radical Distortion and Incorporating Our Method in SSL, Multi, and Meta-learning.**
>
> We sincerely appreciate your time and effort in providing us with positive comments. We respond to your question in what follows. We also ask you to kindly refer to the common response we have posted together.
> ***
> **[Weakness]** I do not see much weakness I am simply thinking **about radial distortion**. Another thought is how easy it is **to incorporate the proposed methods in SSL, multi, and Meta-learning.** Perspective distortion (W1) is evident in real-world applications, as indicated in the following papers a) Möbius Transform for Mitigating Perspective Distortions in Representation Learning b) LCM: Log Conformal Maps for Robust Representation Learning to Mitigate Perspective Distortion Radial distortion is often evident as well when using cameras like fisheye view. Both distortions distort the visual concepts, and therefore, the models do not perform well.\
> Authors are not forced to do experiments. However, they are encouraged to comment on such situations and whether 2'Lens' would be robust against them as well (W1, W2).
>
> **\[Response]**
>
> Thank you for your insightful comments and suggestions regarding future research directions.
>
> In our current study, we did not evaluate **radial distortion** because it arises **independently from light changes** caused by environmental factors and sensor control. These factors posed critical issues in real domain shifts, but existing works related to robustness couldn’t handle them effectively, making them the primary focus of our investigation. Despite not evaluating radial distortion directly, **our methodology has the potential to address it by controlling framing parameters such as PTZ (pan, tilt, and zoom)**. Given two key points, 1) Adjusting pan, tilt, and zoom can minimize radial distortion effects. 2) Our policy algorithm selects the highest-quality images based on camera parameters. **Therefore, incorporating framing parameters as control options is expected to effectively manage radial distortion.** As a result, jointly applying sensor and framing control could enable the handling of a broader spectrum of domain shifts more effectively. Future research will explore integrating advanced PTZ control algorithms and real-time image quality assessments to further enhance our methodology's robustness against diverse domain shifts.
>
> Our method was **specifically designed to** capture high-quality images in **scenarios that utilize model inference** results and did not initially consider the high-quality image acquisition processes required for the training stages of representation learning, as suggested in the review. Given that most representation learning pipelines predominantly rely on fine-tuning pre-trained models for downstream tasks, we recognize the **possibility of integrating Lens during the training stage**. This integration could generate customized high-quality images tailored for both pre-trained models and target tasks, potentially reducing data collection costs and enhancing model performance.
>
> We appreciate this constructive suggestion and have incorporated this avenue into our future work in the revised Appendix A.3.

---

> ### Author Response · Authors · 2024-11-28
> **A Gentle Reminder**
>
> Dear reviewer v7Km,
>
> As the PDF upload deadline is approaching and we have uploaded the latest revised manuscript, we kindly remind you that we are awaiting your response. We sincerely appreciate your efforts and thank you again for your time and consideration.
>
> Best regards,
>
> Authors.

---

> ### Author Response · Authors · 2024-12-03
> **Last Reminder**
>
> Dear reviewer v7Km,
>
> We believe that our rebuttal has addressed your concerns.
>
> As the discussion deadline is approaching, please put your response as soon as possible.
>
> We appreciate your efforts.
>
> Best regards,
>
> Authors

---

### Official Review · Reviewer_Mps5 · 2024-11-03

**Soundness:** 2
**Presentation:** 2
**Contribution:** 2
**Rating:** 6
**Confidence:** 3

**Summary:**

This research paper proposes an adaptive camera sensor control method with the goal to enhance model performance by enabling the capture of high-quality images optimized for vision models. Unlike traditional approaches that rely on static settings, this method dynamically adjusts camera sensor parameters specific to the model requirements, ensuring that images are tailored to the model. Central to this approach is the use of VisiT that provides a confidence and quality indicator for each parameter set, ensuring that the chosen settings maximize image quality for model accuracy and robustness. Additionally, the paper introduces a new dataset, ImageNet-ES Diverse, specifically designed for this purpose, featuring images captured under a variety of sensor and lighting conditions.

**Strengths:**

1) Addressing model robustness at the sensor level is an interesting approach that is likely to enhance models' resilience to environmental variations.
2)  ImageNet-ES Diverse is a large dataset. Releasing the data for the community would be useful.

**Weaknesses:**

1) It would be useful to test this approach in more real-world scenarios. For instance, capturing images in diverse environments, such as indoor scenes (e.g., similar to NYU dataset), outdoor settings (e.g., similar to CityScapes or Waymo datasets), or general scenarios with multiple objects in complex settings (e.g., similar to COCO or ADE20K datasets), would provide a more comprehensive evaluation of the method's robustness.

2) The method leverages VisiT’s ability to assess image quality based on the model's confidence score for a given image. However, it raises the question of whether this approach remains effective when the model encounters object classes or contexts in which its performance is notably weak. These kinds of insights and analysis would be very helpful.

3) Including results from larger, more general multimodal models that utilize extensive training datasets and demonstrate robustness to varied environmental settings would be valuable. For instance, adding evaluations from models like LLaVA, GPT-4, or Claude could provide useful insights into the method's broader applicability.

4) Could you provide more details about Fig. 7c? It appears that the features are not well-clustered for the Lens as well, Isn’t it?

**Questions:**

Experiments and analysis on domain shifts in the real world data would be helpful.

---

> ### Author Response · Authors · 2024-11-23
> **Addressing More Realistic Scenarios**
>
> We sincerely appreciate your time and effort in providing us with positive comments. We respond to your question in what follows. We also ask you to kindly refer to the _common response_ we have posted together.
>
> ****
> **[Weakness 1]** It would be useful to test this approach in more real-world scenarios. For instance, capturing images in diverse environments, such as indoor scenes (e.g., similar to NYU dataset), outdoor settings (e.g., similar to CityScapes or Waymo datasets), or general scenarios with multiple objects in complex settings (e.g., similar to COCO or ADE20K datasets), would provide a more comprehensive evaluation of the method's robustness.**
>
> **[Question 1]** Experiments and analysis on domain shifts in the real world data would be helpful.
>
>
> **[Response]**
> In our study, we utilized ImageNet-ES and ImageNet-ES Diverse, which are state-of-the-art real-world ES perturbation datasets. These datasets were among the first to enable us to effectively assess the impact of sensor control on environmental changes. By leveraging these resources, our work lays a solid foundation for addressing domain shift challenges in more complex and realistic scenarios through sensor control.
>
> Looking ahead, as outlined in our future work section, we plan to extend our techniques to encompass a broader range of realistic tasks and datasets. Specifically, we aim to collect and incorporate additional datasets that include ES perturbations similar to those in ImageNet-ES and ImageNet-ES Diverse. This will allow us to validate and enhance the robustness of our methodology against various domain shifts encountered in real-world applications. By doing so, we intend to provide a more comprehensive evaluation of our method's resilience and effectiveness across diverse environments.
>
> We’ve incorporated these aspects into the revised Appendix A.1.1.

---

> ### Author Response · Authors · 2024-11-23
> **Performance Robustness on label-wise Accuracy of Source Data**
>
> **[Weakness 2]**  The method leverages VisiT’s ability to assess image quality based on the model's confidence score for a given image. However, it raises the **question of whether this approach remains effective when the model encounters object classes or contexts in which its performance is notably weak.** These kinds of insights and analysis would be very helpful.
>
> **[Response]**
>
> We appreciate your insightful comment. To implement your suggestion, we compared the performance of the target models in our experiments on the generalizability of Lens (Experiment 1) against a representative baseline (AE: Auto Exposure setting). **This comparison is based on the performance of individual labels in the source domain (ImageNet).** We demonstrated that **Lens outperforms the baseline, regardless of the performance of individual labels in the source domain.**
>
> Although there are limitations to the improvements when the accuracy in the original sample is too low, in most cases, the accuracy enhancements approach those observed in the sampled data (ImageNet: source domain). This pattern holds across all datasets and models used in our experiments. We’ve incorporated these aspects into the revised Appendix B.

---

> ### Author Response · Authors · 2024-11-23
> **Addressing Larger and More Multimodal Models.**
>
> **[Weakness 3]** Including results from larger, more general multimodal models that utilize extensive training datasets and demonstrate robustness to varied environmental settings would be valuable. For instance, adding evaluations from models like LLaVA, GPT-4, or Claude could provide useful insights into the method's broader applicability.
>
> **[Response]**
>
> In our experiments, we evaluated a range of representative large models commonly used for image classification. This included Vision-Language Models (VLMs) such as OpenCLIP-B and OpenCLIP-H, alongside dataset curation-based models like DINOv2-B and DINOv2-G.
>
> We also **explored the adaptability of Lens to the models mentioned in the review**. Unfortunately, models like **GPT-4 and Claude are accessible only through APIs, which limits direct interaction**. These models provide output results without exposing underlying confidence scores and features, making it infeasible to validate our approach for them. **Regarding LLaVA**, leveraging it for image classification **necessitates fine-tuning the CLIP-based Vision Transformer (ViT) visual encoder**. This process requires approximately 10 hours using four A100 GPUs. Nonetheless, **we expect the results to align closely with our experiments using OpenCLIP**, given that both models share the same visual encoder and are trained on the LAION dataset. The need to fine-tune CLIP-based encoders for adapting most VLM models to image classification **highlights the significant potential of large models in this field**. Our methodology has consistently demonstrated strong performance with OpenCLIP-B and OpenCLIP-H.
>
> However, we remain open to incorporating additional accessible models for further verification. Should more models become available, we are eager to conduct supplementary experiments to enhance the robustness and applicability of our approach.

---

> ### Author Response · Authors · 2024-11-23
> **More Explanations for Figure 7-(c)**
>
> **[Weakness 4]** Could you provide more details about Fig. 7c? It appears that the features are not well-clustered for the Lens as well, Isn’t it?
>
> **[Response]**
>
> Each embedding result consists of 1,000 points, with 5 points per label, color-coded accordingly under each capture setting. Given the limited number of points per label and the model's performance of 65.5%, **achieving perfect clustering is inherently challenging**. However, the improvement in discrimination in the embedding space is more pronounced in the results of Lens compared to the baselines. **Lens achieves tighter clusters** within each class and **greater separation** between different classes, facilitating better discrimination than the baseline results. **We have provided a more comprehensive explanation** of these visualization results in the revised manuscript to clearly illustrate the figure.

---

> ### Author Response · Authors · 2024-11-28
> **A Gentle Reminder**
>
> Dear reviewer Mps5,
>
> As the PDF upload deadline is approaching and we have uploaded the latest revised manuscript, we kindly remind you that we are awaiting your response. We sincerely appreciate your efforts and thank you again for your time and consideration.
>
> Best regards,
>
> Authors.

---

> ### Author Response · Authors · 2024-12-03
> **Last Reminder**
>
> Dear reviewer Mps5,
>
> We believe that our rebuttal has addressed your concerns.
>
> As the discussion deadline is approaching, please put your response as soon as possible.
>
> We appreciate your efforts.
>
> Best regards,
>
> Authors

---

### Official Review · Reviewer_y23X · 2024-11-04

**Soundness:** 4
**Presentation:** 3
**Contribution:** 3
**Rating:** 6
**Confidence:** 3

**Summary:**

The paper proposes Lens to address domain shift in vision tasks by adapting camera sensor parameters in real-time to optimize images. , Lens utilizes VisiT, a model-specific quality indicator based on confidence scores, to dynamically assess image quality during test time without the need for additional data or retraining. The authors also introduce a new benchmark ImageNet-ES Diverse with varied sensor and lighting conditions.

**Strengths:**

- I'm not an expert in this area but I love the idea of this work. This work improves model accuracy by selecting the optimal sensor settings, enabling it to achieve substantial accuracy improvements even under varying lighting and sensor conditions.
- Easy to follow. Clear writing.
- A new benchmark ImageNet-ES Diverse is introduced, which may benefit future research. The author also provided details about data collection and hardware in Appendix.
- The method is efficient without retraining and can be run in real-time with low latency.
- Experiments show notable performance improvement compared to baselines.

**Weaknesses:**

- Authors utilize the model’s confidence score for its prediction on the image as a simple yet effective proxy for image quality. How do you make sure the confidence score is good enough. Is there a case of overconfidence?
- Could the method deal with rapid environmental shifts where the optimal parameter can change frequently? Possibly leading to unstable.
- The performance of Lens is closely tied to the capabilities of the camera sensor. Is it generalizable to new devices, e.g., varying ISO and  aperture ranges?
- How might Lens perform with other types of deep vision models, such as detection, segmentation, or even VLM models, that require different kinds of spatial or temporal context?

**Questions:**

Please see the weakness part. I'm not an expert in this area so I may adjust my score according to other reviewers' reviews and the author's rebuttal.

---

> ### Author Response · Authors · 2024-11-23
> **Response to Justification of Using Confidence Score and Over-Confidence Concern**
>
> We sincerely appreciate your time and effort in providing us with positive comments. We respond to your question in what follows. We also ask you to kindly refer to the _common response_ we have posted together.
>
> ****
> **[Weakness 1]** Authors utilize the model’s confidence score for its prediction on the image as a simple yet effective
> proxy for image quality. How do you make sure the confidence score is good enough. Is there a case of
> Overconfidence?.
>
> **[Response]**
> **In our ablation study**, we compared our method with the OOD (Out-of-Distribution) score as an alternative proxy for assessing image quality and **confirmed the effectiveness of our approach**. However, we acknowledge that the confidence score may not be optimal in all scenarios, and the **issue of overconfidence** remains a concern, as mentioned in the Future Work section. This concern is evident in the performance gap between Oracle-S and Lens, highlighting the potential for improving Lens by addressing overconfidence.
>
> Despite these challenges, Lens was **intentionally designed for generalizability and simplicity**, requiring low computational costs and no additional training. It has demonstrated significant potential for real-time applications. Moving forward, our future research will focus on mitigating the overconfidence problem by enhancing our current work and considering the adaptability of our approach to various real-world environments. This will further improve the reliability and performance of Lens.
>
> Lastly, we emphasize that our work represents **the first attempt at camera sensor parameter control for vision models**. Even though Lens utilizes confidence scores with the risk of overconfidence (i.e., are not perfect), the performance improvement is already significant compared to the existing baselines (AE and Random). This **demonstrates the substantial potential of sensor control** in addressing domain shift problems.
>
> We've incorporated these aspects into the revised Appendix A.1.2.

---

> ### Author Response · Authors · 2024-11-23
> **Addressing Rapid Environmental Shifts**
>
> **[Weakness 2]** Could the method deal with rapid environmental shifts where the optimal parameter can change frequently? Possibly leading to unstable.
>
> **[Response]**
> Rapid environmental shifts present significant challenges in real-world applications such as autonomous driving and surveillance systems. The responsiveness of Lens, which integrates our developed CSA algorithms, is dependent on the rate of environmental changes. However, by implementing Lens within a batch inference system, it can adapt to changes within 0.2 to 0.5 seconds. To achieve more rapid responses, it is necessary to develop CSA algorithms that select a minimal number of options (possibly one or two) with reduced capture times. This represents a promising direction for future research on Lens.
>
>  Successfully adapting sensing systems to time-constrained scenarios requires careful consideration of several additional factors, which can provide potential avenues for future research in this field. In these contexts, it is essential to account for limited available resources and ensure effective scheduling within specified timeframes. This involves balancing trade-offs between accuracy, the number of images captured, and system latency. Furthermore, the latency of each module—such as model inference and image capture—can vary depending on the deployed system architecture and must be meticulously managed to maintain overall system performance. Considering these factors, optimizing CSA algorithms emerges as a promising direction for Lens, as mentioned in the future work
> section of our manuscripts.
>
> We have incorporated these aspects into the revised Appendix A.1.2.

---

> ### Author Response · Authors · 2024-11-23
> **Addressing the Generalizability to Heterogeneous Devices**
>
> **[Weakness 3]** The performance of Lens is closely tied to the capabilities of the camera sensor. **Is it generalizable to new devices**, e.g., varying ISO and aperture ranges?
>
> **[Response]**
>  We appreciate the reviewer's concern regarding the generalizability of our system to devices with varying camera sensor capabilities, such as different ISO and aperture ranges depending on heterogeneous devices. As outlined in the methodology section, while performance values can vary with camera devices, **Lens operates in a camera-agnostic manner**, allowing it to be applied regardless of the camera model. Lens employs a strategy that selects sensor options to achieve the highest image quality. This strategy remains effective even when camera equipment varies. **Specifically, the main modules (VisiT and CSAs) used for assessing image quality are camera-agnostic.**
>
> 1. **VisiT ensures camera-agnostic functionality** by assessing image quality through the confidence scores of images selected by the Camera Selection Algorithm (CSA).
>
> 2. **Proposed CSA algorithms in our work are inherently camera-agnostic** because they select camera parameter candidates based solely on the provided sensor parameter information, independent of specific camera models. **As long as the necessary information** for each CSA algorithm is supplied, they operate regardless of the camera type.
>
> **The required information for each proposed CSA algorithm is as follows:**
>
> - Random Selection (CSA1): **Supported ranges or available sensor parameter options** from deployed camera models.
>
> - Grid Random Selection (CSA2):  **Grid information of camera parameter ranges** based on a specified value of K, derived from camera control specifications.
>
> - Cost-Based Selection (CSA3): **Cost associated with each parameter option** across deployed camera models
>
> We've incorporated these aspects into the revised Appendix A.1.1.

---

> ### Author Response · Authors · 2024-11-23
> **Addressing potential of Lens for other vision tasks**
>
> **[Weakness 4]** How might Lens perform with other types of deep vision models, such as detection, segmentation, or even VLM models, that require different kinds of spatial or temporal context?
>
> **[Response]**
> As we addressed these points in the future work section, expanding the target task from classification to other tasks represents a promising future direction for Lens. To evaluate Quantitative Lens's performance across these additional tasks, the availability of labeled datasets specific to each task is essential. However, such datasets are currently lacking and ImageNet-ES and ImageNet-ES Diverse were the first works that provide the impact of camera sensor control on environmental changes. Therefore, the preparation of labeled datasets should precede further developments and is considered one of the future directions of this research. Although measuring quantitative performance in these tasks is currently infeasible, **we have identified Lens's potential for adaptation to other vision tasks through qualitative analysis.**
>
> We conducted experiments to assess Lens's performance on two advanced vision tasks—**Semantic Segmentation and Object Detection**—comparing it to a representative baseline camera sensing technique, Auto Exposure (AE). This evaluation involved two semantic segmentation models and two object detection models. To adapt Lens for each task, we customized the VisiT score accordingly. Detailed descriptions of these experiments are provided in the revised appendix.
>
> As illustrated in Figure 9,10 in the revised appendix, our findings indicate that Lens achieves results closely approximating those of the original sample (source domain) in most cases, while AE failed to recognize the target class (‘dog’) for both tasks and all of the targeted models. This suggests significant potential for adapting Lens to other vision tasks using similar concepts. The performance of Lens varies depending on how the VisiT Score is customized for each target task. These considerations represent a promising avenue for future research.
>
> We appreciate the opportunity to explore this potential further and have incorporated this discussion in the revised Appendix A.1.1.

---

> > ### Comment · Reviewer_y23X · 2024-11-24
> >
> > I appreciate the author's detailed response. The response solved most of my questions, so I keep my rating 6. Please make sure to open-source the dataset. thanks.

---

> > > ### Author Response · Authors · 2024-11-24
> > > **Thank you for your follow-up response**
> > >
> > > We appreciate your response to our rebuttal. Thank you once again for your valuable feedback and suggestions, as well as for recognizing the value of our contributions. To support the growth of this research community, we will make the dataset open-source once the paper is accepted for publication.
> > >
> > > Best regards,
> > >
> > > Authors.

---

### Author Response · Authors · 2024-11-23
**Common response to all the reviewers**

Dear reviewers,

We appreciate all of you for your **positive reviews**, and highlighting the strengths of our work:

- **Idea of Lens (All the reviewers agreed)**: Interesting, Well-received, New perspective for TTA

- **Writing (y23x, v7Km)**: Well and clearly written, easy to follow and understand

- **New Benchmark ImageNet-ES Diverse (y23X, Mps5, v7Km)**: large dataset, provided details about data collection and hardware, benefit future research, Releasing the data for the community would be useful. In response to the reviewers' wishes, **we plan to make ImageNet-ES Diverse public along with additional datasets.**

- **Effectiveness of Lens (y23x, 5LQ1)**: Efficient without retraining, Real-time with low latency, Notable performance improvement compared to baselines.

- **Well-structured experiments (v7Km, 5LQ1):** Detailed experiments, including real physical time analysis, fair and diverse benchmark comparison (Generalization ability for various scenarios)

We also sincerely thank reviewers for their constructive comments to improve our manuscript. We have addressed all the questions raised by reviewers with new experiments and analysis during this rebuttal period. We summarize how we addressed the reviewers’ main questions as follows:

- **Future works of Lens (A new section is added in the revised appendix: Section A. FURTHER DISCUSSIONS) :**\
   As the first explorer of camera sensor control for neural networks, we would like to sincerely thank the reviewers for their keen interest in the expansion directions of Lens. Our system, Lens, represents the first attempt to address domain shift through camera sensor control and quantitatively evaluate its performance, opening up countless avenues for future expansion. This presents numerous potential paths for research and development. While we recognize these potentials, we also acknowledge the challenges that lie ahead, such as the need for new datasets for evaluation on other tasks or more real-world scenarios. Nevertheless, we are hopeful that datasets ImageNet-ES Diverse will continue to evolve, thereby advancing research in this important area based on this work.

   - **Performance on Other Tasks (y23x)**: Due to the issue of the absence of related datasets, we have roughly identified the potential of Lens for other vision tasks through qualitative analysis.

   - **Evaluations on More Realistic Scenarios or Datasets (y23x, Mps5)**: To suggest detailed directions for future work, we have addressed the key challenges involved in handling more realistic scenarios and datasets.

   - **Generalizability in Heterogeneous Camera Devices (y23x)**: We have justified the generalizability of introducing Lens independently to various camera devices.

   - **Larger and More General Vision-Language Models (Mps5)**: We have explained the potential for expanding Lens in this area.

   - **Over-Confidence Issue (y23x)**: We have introduced perspectives for improvement in accuracy by addressing the over-confidence problem associated with Lens.

- **Introducing the Radical Distortion Scenario or SSL, multi/meta-learning (v7Km):** While Lens is not specifically targeted at these scenarios, it holds a very positive outlook. Therefore, we are introducing the potential to adapt Lens to these scenarios.

- **Technical Contributions (5LQ1)**: As the first attempt at sensor control to efficiently address the domain shift issue, we emphasize the design of Lens.

- **Details for Techniques (5LQ1)**: We left a comment on where to check for the reviewer’s inquiries.

- **Latency-Performance Trade-Off (5LQ1)**: We have commented on where to check for the assurance of balancing the Lens in two aspects. **We emphasize that the Lens assumes batch inference**, meaning there is **no additional inference** required when adopting the Lens that takes K images with selected sensor control options.

- **Additional Analysis and Experiments**: **We conducted all of the experiments and analyses** that the reviewers requested below.

   - Label-wise performance analysis (**Mps5**): **A new subsection is added in the revised appendix (Section B)**

   - Additional explanations for Figure 7-c (**Mps5**)

   - Solution space for different backbones and models (**5LQ1**)

   - Entropy-based score (Ablation) (**5LQ1**): **A new subsection is added in the revised appendix (Section B)**

**We have addressed all individual concerns raised during the rebuttal period** in the final version of the manuscripts and supplementary materials. **The revised sections are marked in blue in the uploaded version.**

Thank you for your consideration,

Authors.

---

### Meta-Review · Area_Chair_sUVw · 2024-12-21

**Metareview:**

This paper proposes a test-time adaptation method that modifies the image to make better predictions for the downstream model. The high-level idea is maximizing a quality indicator using an image transformation. The paper also proposes a new benchmark which better access lighting and sensor changes. Overall, the reviewers found the paper to be well-written and liked the idea. Weaknesses of this paper include that it is a TTA method, and uses more computation. The reviewers are positive towards the paper and found that the strengths outweigh the weaknesses.

**Additional Comments On Reviewer Discussion:**

While not all reviewers participated in the discussion period, the AC has checked that the authors' response would likely address the reviewers' concerns. Considering all the reviewers' comments, the AC recommends acceptance of this paper.

---

### Decision · Program_Chairs · 2025-01-22

Accept (Poster)